# Four ZIPs contribute to Zn, Fe, Cu and Mn acquisition at the outer root domain

Kevin Robe*, Linnka Lefebvre-Legendre, Fabienne Cleard, Marie Barberon*

Department of Plant Sciences, University of Geneva, Geneva, Switzerland

* kevin.robe@unige.ch (KR); marie.barberon@unige.ch (MB)

## Abstract

Zinc (Zn), an essential micronutrient, plays a crucial role in plant development. However, the specific transporters involved in Zn uptake from the soil remain unclear in dicotyledonous plants. Using promoter-reporter lines in *Arabidopsis thaliana*, we identified several *ZIP* (*Zn-regulated transporter, Iron-regulated transporter (IRT)-like Protein*) family members that are expressed in the epidermis and potentially involved in Zn acquisition from the outer root domain. ZIP2, ZIP3, ZIP5 and ZIP8 predominantly localize to the plasma membrane of epidermal and cortical cells, supporting their potential roles in metal uptake from the soil. Through physiology studies, ionomic profiling and genetic analysis, we determined that ZIP3 and ZIP5 are contributors to Zn acquisition, while ZIP2 and ZIP8 are primarily involved in copper (Cu) and iron (Fe) acquisition respectively. Notably, ZIP3 and ZIP8 exhibit outer polarity in root epidermal cells, similar to IRT1, in agreement with expectations of transporter polarity in mineral acquisition. These findings provide new insights into the mechanisms of metal uptake in plant roots and offer potential strategies for biofortification to enhance metal content in plants.

## Author's Summary

Plants absorb essential mineral nutrients like zinc (Zn), iron (Fe), and copper (Cu) from the soil through their roots, but the specific proteins that mediate this process are not fully known. In this study, we investigated a group of transporter proteins called ZIPs in the model plant *Arabidopsis thaliana* to determine their role in metal uptake. We focused on ZIP2, ZIP3, ZIP5, and ZIP8, which are located at the root periphery, precisely where metal uptake from the soil occurs. By combining gene expression studies, microscopy, ionomics, and mutant analysis, we showed that ZIP3 and ZIP5 contribute to Zn uptake, while ZIP2 and ZIP8 help plants acquire Cu and Fe, respectively. Furthermore, we discovered that some of these transporters are polarized toward the outer surface of root cells,

**Data availability statement:** The sequence data used in this article can be found in the GenBank/EMBL databases under the following numbers: IRT1 (AT4G19690), IRT2 (AT4G19680), IRT3 (AT1G60960), ZIP1 (AT3G12750), ZIP2 (AT5G59520), ZIP3 (AT2G32270), ZIP4 (AT1G10970), ZIP5 (AT1G05300), ZIP6 (AT2G30080), ZIP7 (AT2G04032), ZIP8 (AT5G45105), ZIP9 (AT4G33020), ZIP10 (AT1G31260), ZIP11 (AT1G55910), ZIP12 (AT5G62160). All other relevant data are in the manuscript and its supporting information files.

**Funding:** This work was supported by the Swiss National Science Foundation (PCEGP3_187007) and the Sandoz Family Monique De Meuron philanthropic foundation's program for academic promotion (https://www.fpfs.ch/pages/la-fondation) awarded to MB, and by the state of Geneva. The funders had no role in study design, data collection and analysis, decision to publish, or preparation of the manuscript.

**Competing interests:** The authors have declared that no competing interests exist.

a trait consistent with their role in importing metals from the environment. Our findings clarify how plants acquire key micronutrients and highlight targets for improving nutrient content in crops through biofortification or breeding.

## Introduction

Plants require at least fourteen essential mineral nutrients for proper growth and development [1]. Among these, the transition metal iron (Fe) has been extensively studied, largely due to the pronounced phenotypic consequences of Fe deficiency. In contrast, other transition metals such as manganese (Mn), copper (Cu) and zinc (Zn), while equally critical for various biological processes, have received comparatively less attention. Zinc, in particular, plays a crucial role as a structural and catalytic cofactor in numerous proteins, including RNA polymerase, superoxide dismutase, and alcohol dehydrogenase, and is present in nearly 10% of eukaryotic proteomes [2–4]. However, in natural soils, Zn is only sparingly soluble, as most of it is bound to minerals such as sphalerite (ZnS), franklinite ($ZnFe_2O_4$), Zn-containing magnetite ($[Fe,Zn]Fe_2O_4$), willemite ($Zn_2SiO_4$), hemimorphite ($Zn_4Si_2O_7[OH]_2 \cdot H_2O$), and zincite (ZnO) [5]. Among these, sphalerite is a primary source of soluble $Zn^{2+}$ in the soil solution [6]. Soil pH is a major factor influencing Zn availability and speciation, with its solubility decreasing as pH increases. Consequently, plants frequently experience Zn deficiency in alkaline soils, which constitute approximately 30% of cultivated land globally [7]. Both Zn deficiency and excess can result in significant growth defects and yield reduction in the field.

Several transporter families play key roles in Zn transport within plants, such as the Zrt-/Irt-related Proteins (ZIP) family, the Yellow Stripe Like (YSL) family, the Heavy Metal ATPase (HMA) family and the Metal Tolerance Proteins (MTP) family [2,3,8,9]. In Arabidopsis, for example, Zn efflux into the xylem is primarily mediated by AtHMA2 and AtHMA4, which are predominantly expressed in pericycle cells, while vacuolar Zn storage depends on AtMTP1 [10–12]. In monocotyledonous plants, Zn can be taken up either as divalent cation ($Zn^{2+}$) or in a chelated form bound to phytosiderophores. However, it has been reported that Zn uptake mediated by ZmYS1 from phytosiderophores in maize is much lower than uptake from $ZnSO_4$, suggesting that Zn is primarily acquired as a cation [13]. Further supporting this, the accumulation of the stable isotope $^{67}Zn^{2+}$ was nearly abolished in the *oszip5oszip9* double mutant of *Oryza sativa*, indicating that OsZIP5 and OsZIP9 are the primary transporters responsible for Zn acquisition in rice. Both transporters are expressed in the root epidermis and are localized to the plasma membrane (PM) [14–16]. These findings marked the first demonstration of Zn transporters playing a major role in Zn acquisition in monocotyledonous plants.

While the molecular components involved in Zn storage and transport are well characterized, the mechanisms underlying Zn uptake by roots in dicotyledonous plants, including Arabidopsis, remain poorly understood. Several transporters were proposed to facilitate root Zn uptake, including AtIRT1, AtIRT3 and other members

of the ZIP family [17–19]. In Arabidopsis, there are 15 ZIP family members, including 3 IRTs and 12 ZIPs. Yeast strains defective in specific metal transport pathways, such as *zrt1zrt2* for Zn, *fet3fet4* for Fe, *smf1* for Mn or *ctr1* for Cu, were widely used to test ZIP metal transport activity [18,20–22]. ZIP transporters can transport a range of divalent cations when expressed in yeast, including $Zn^{2+}$, $Fe^{2+}$, $Mn^{2+}$, $Cu^{2+}$, and cadmium ($Cd^{2+}$) [23]. Although AtIRT3 overexpression results in Zn and Fe accumulation, indicating a possible role in Zn uptake, recent evidences suggest that AtIRT3 primarily functions redundantly in Zn distribution and translocation from root to shoot, without a direct role in Zn uptake from soil [17,24]. In contrast, AtIRT1 can transport Zn and other divalent metals in heterologous systems [19,21,25,26], but its role *in planta* for Zn uptake appears to be restricted to specific conditions, such as Fe deficiency when its expression is induced. Moreover, recent findings show that Fe deficiency-induced Zn accumulation is independent of AtIRT1 [27]. Several other ZIPs, including AtZIP1, AtZIP2, AtZIP3, AtZIP4, AtZIP5, AtZIP6, AtZIP7, AtZIP9, AtZIP11, and AtZIP12 were shown to complement yeast mutants defective in Zn uptake, suggesting that they may contribute to Zn transport in plants [18,20,28]. While many studies have reported Zn transport in heterologous systems, *in planta* characterization of ZIP transporters remains limited. For instance, a slight increase in shoot Zn content was observed in the *zip5* mutant compared to the wild type (WT), but no significant difference was detected in the *zip6* mutant [29]. The *zip2* mutant exhibited Zn overaccumulation in roots, while *zip1* showed no difference in Zn content relative to WT [18]. In contrast, a reduced Zn translocation from root to shoot was reported in the quadruple *irt3zip4zip6zip9* mutant [24]. Promoter activity analysis suggests that these four ZIP transporters are involved in Zn transport to the stele, but not directly in Zn uptake from the soil. Very recently, in parallel to our study, the redundant roles of ZIP3 and ZIP5 in Zn uptake and the response to Zn deficiency were demonstrated [30].

In our search for transporters involved in Zn uptake in the dicotyledonous Arabidopsis, we first assessed the expression of the entire *ZIP* family in roots under both Zn-sufficient and Zn-deficient conditions and investigated their promoter activity in roots. Based on these data, we selected ZIP transporters expressed at the root periphery as putative candidates for Zn acquisition and characterized their functional roles. Our *in planta* evidences, derived from mutants and overexpressing lines, indicate that ZIP2 and ZIP8 contribute to Cu and Fe acquisition, respectively, while ZIP3 and ZIP5 are involved in Zn acquisition. Consistent with their functions, ZIP2, ZIP3, ZIP5 and ZIP8 localize at the PM and endomembrane compartments of epidermal and cortical cells. The polarity of epidermal ZIP3 and ZIP8 towards the outer root domain (facing the soil) provides circumstantial support for their significance in Zn and Fe acquisition from the soil and underscores the importance of transporter polarity in their functional roles. Overall, this study highlights the value of re-evaluating transporter localization to uncover new functions. The identification of ZIP3 and ZIP5 as key transporters involved in Zn acquisition from the soil marks a significant advancement in our understanding of mineral nutrition, as Zn was one of the last essential elements for which transporters involved in acquisition had not been unequivocally identified in dicotyledonous plants.

## Results

### Physiological effect of zinc deficiency in *Arabidopsis thaliana*

To investigate the effect of Zn deficiency on plant physiology, we cultivated WT plants in Murashige and Skoog-based zinc-sufficient (15 µM Zn; +Zn) and deficient (0 µM Zn; -Zn) media. After three weeks of growth on horizontal plates under -Zn conditions, the plants showed stunted growth and chlorosis, resulting in a significant reduction in both fresh weight (FW) and chlorophyll content compared to those grown in +Zn conditions (S1A–C Fig). As anticipated, the diminished chlorophyll content correlated with a decrease in the maximum quantum yield of photosystem II (PSII, $F_v/F_m$), which reflects the potential efficiency of PSII, further confirming the impact of Zn-deficiency on photosynthetic performance (S1D Fig). While mineral deficiencies often induce significant compensatory changes at the whole plant level, the specific effect of Zn deficiency on root and shoot ionome profiles have not been extensively documented. To address this gap, we

conducted an ICP-OES (Inductively Coupled Plasma – Optical Emission Spectrometry) analysis of mineral content in the roots and shoots of 3-week-old plants grown in +Zn or -Zn conditions. As expected, Zn was nearly undetectable in both roots and shoots of WT plants under -Zn conditions (S1E and S1F Fig and S1 Table). Moreover, we observed significant reductions in the content of potassium (K), magnesium (Mg), Mn, sodium (Na), strontium (Sr), calcium (Ca), and Cu in the shoots of plants grown in -Zn compared to those grown in +Zn. Conversely, the levels of Fe and molybdenum (Mo) remained largely unaffected by Zn deficiency in shoots. Notably, Zn-deficient roots accumulated over three times more Mn than control roots, suggesting that the transport systems induced under Zn deficiency may also facilitate Mn uptake. Additionally, we found that Mg, Sr, Fe and K levels were significantly reduced in the roots under -Zn conditions while the contents of Ca, Cu, Mo and Na remained unchanged. We also assessed the translocation of minerals from roots to shoots under both +Zn and -Zn conditions. The root-to-shoot translocation of Ca, Cu, K, Mn, Mo, Na and Sr decreased under -Zn condition. In contrast, the translocation of Fe and Zn increased under -Zn conditions. The translocation of Mg from roots to shoots remains consistent between +Zn and -Zn conditions (S1G Fig). Overall, these findings establish conditions under which Zn deficiency significantly affects plant development, physiology and overall mineral accumulation. Furthermore, they suggest that compensatory mechanisms may be at play for other minerals in -Zn conditions.

## Spatio-temporal expression analysis of the *AtZIP* family reveals potential candidates for metal acquisition from the soil

To gain insight into the transcriptional response of plants under Zn-deficient conditions and identify candidates for Zn uptake from the soil, we first assessed the expression of the entire *ZIP* family in WT plants grown for 1 week in either +Zn or -Zn conditions (Fig 1A). The expressions of *IRT1*, *ZIP6*, *ZIP7*, *ZIP10* and *ZIP11* were not significantly altered by Zn deficiency (Figs 1B and S2A). In contrast, we observed increased mRNA accumulation of *IRT3*, *ZIP1*, *ZIP3*, ZIP4, *ZIP5*, *ZIP9* and *ZIP12* in WT roots grown under Zn deficiency compared to control conditions, while *IRT2, ZIP2* and *ZIP8* showed decreased expression, similarly to what previous studies reported [30–33]. These results suggest that the upregulated transporters may serve as potential candidates for Zn uptake in conditions of limited Zn availability. Notably, *ZIP2* was the only member of the *ZIP* family that exhibited high expression under control conditions (S2A Fig). This observation aligns with previous reports of high *ZIP2* expression in control conditions [18].

Anticipating that transporters involved in Zn uptake from the soil would be expressed at the root periphery, we next examined the spatial expression pattern of the 15 ZIP transporters using promoter-reporter lines. We investigated promoter activity in the root tip (zone I), the fully elongated zone (zone II) and the differentiated root (zone III) in control conditions (i.e.,; containing 15 µM Zn). The promoters were used to express a fluorescent reporter, NLS-3xmVenus, in 5-day-old seedlings (Fig 1C). Promoters of *IRT1*, *IRT2*, *ZIP2*, *ZIP3*, *ZIP5* and *ZIP8* displayed activity primarily at the root periphery in zone II and III (*i.e.,* mainly in epidermal and cortical tissues), while the promoter of *ZIP7* was predominantly active in central root tissues such as the endodermis and the stele. Z*IP11* also showed activity in central root tissues but was mostly active in cortical cells. *IRT3*, *ZIP1* and *ZIP10* exhibited almost exclusively endodermal promoter activity (with *ZIP10* being active only in zone III), while *ZIP6* promoter activity was primarily detected in the pericycle in both zones II and III. Promoter activity for *ZIP4*, *ZIP9* and *ZIP12* was minimal under control conditions (as indicated by the absence of nuclear signal; the faint green signal likely corresponds to background fluorescence), with *ZIP12* displaying extremely low signals at the root periphery and in the inner tissue of the zone I. These observations suggest that ZIP2, ZIP3, ZIP5 and ZIP8 may be involved in metal uptake from the soil at the root periphery, similarly to the well-characterized IRT1 in this condition. Given *IRT3*, *ZIP1* and *ZIP10* endodermal promoter activity in control conditions, these three transporters might play a role in metal uptake and transport within the endodermis. Interestingly, six *ZIP* promoters (*IRT2*, *IRT3*, *ZIP1*, *ZIP2*, *ZIP3* and *ZIP7*) showed activity in cells at the root periphery in zone I, indicating their potential role in mediating the transport of divalent transition metal ions in the root tip. Having observed that zinc deficiency led to a higher expression for seven ZIP transporters (Fig 1B), we then investigated their respective promoter activity in -Zn conditions in zone III of

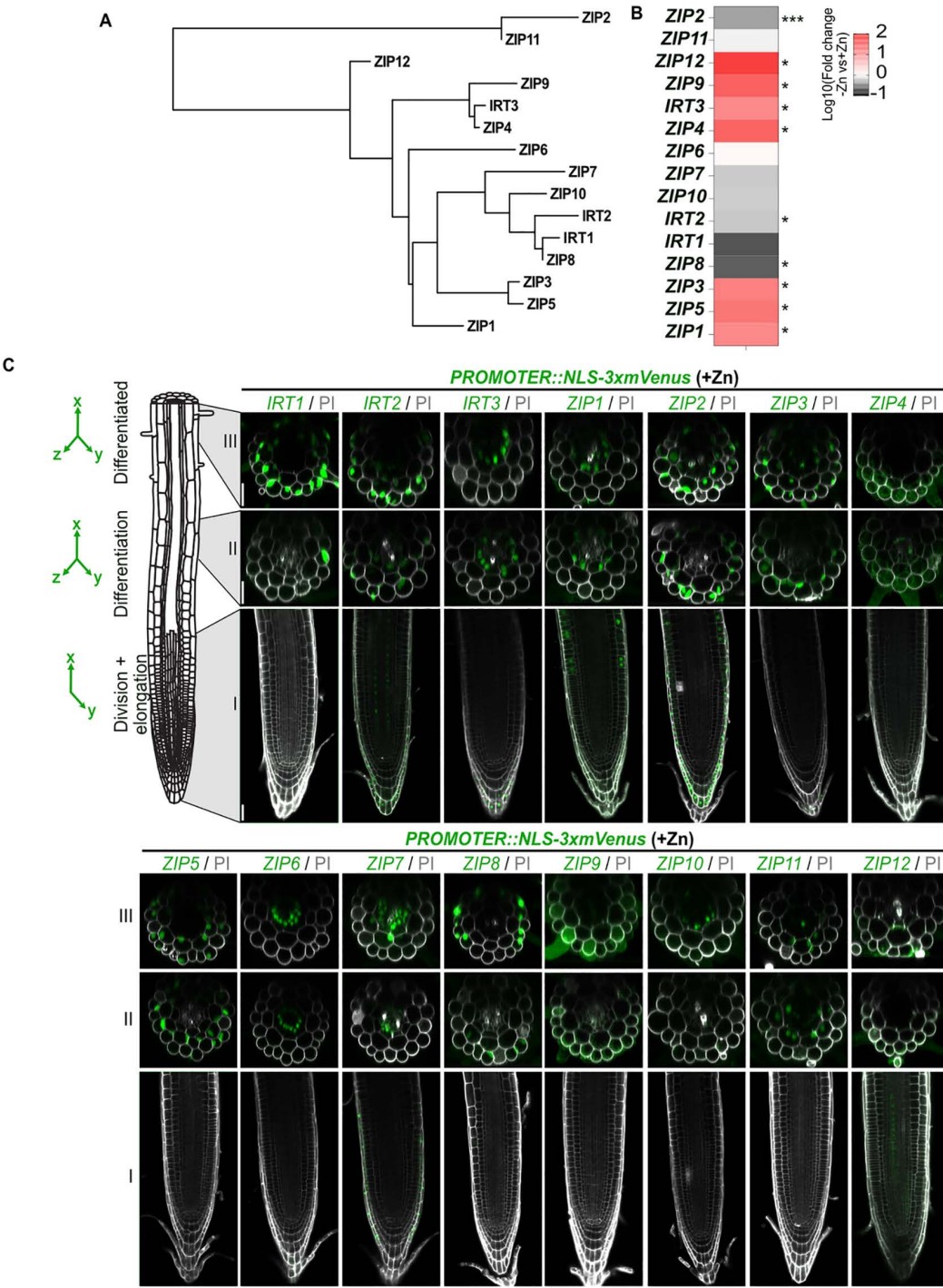

**Fig 1. Spatio-temporal expression of the *AtZIP* family in roots.** **(A)** Phylogenetic tree of the 15 members of AtZIP family in *Arabidopsis thaliana*. The tree was generated using protein sequences with Phylogeny.fr. **(B)** Relative transcript levels of the fifteen *ZIP* members in WT plant roots grown 1 week under control conditions (+Zn) or Zn-deficient conditions (-Zn). Results are presented as a heatmap, highlighting fold changes compared to control conditions (+Zn). Numeric values are provided in S2A Fig. Data represent 4 biological replicates, with each replicate being a pool of at least 40 roots. Statistical differences were determined using Student's *t*-test or Mann Whitney test with significance levels indicated (*$P < 0.05$; **$P < 0.01$; ***$P < 0.0001$). **(C)** Representative images of *ZIP::NLS-3xmVenus* expression pattern from five-day-old plants grown in control conditions. Images taken in different root zones: zone I (root tip, lower pictures), zone II (early differentiated zone, middle pictures), and zone III (mature zone, upper pictures). Nuclear-localized

mVenus signal is shown in green while PI used to stain the cell wall is shown gray. Images were captured with identical settings for each line along the root, but settings varied between lines due to differences in promoter activity among *ZIPs*. For zone I, a singe focal plant for mVenus and PI is displayed (XY), while for zone II and III, mVenus signal is presented as a maximum projection of Z-stacks (XYZ), overlaid with a single orthogonal view of PI extracted from the Z-stacks. Images correspond to T1 plants; at least 5 independent T1 plants were imaged. The signal was confirmed in T2 plants. Scale bars: 25 μm for all images.

the root (S2B Fig). The localization of promoter activities for *ZIP3* and *ZIP5* were similar to the ones observed in control conditions (Fig 1C). However, zinc deficiency led to an expansion to additional cell layers for *ZIP1* promoter (in the stele), *ZIP4* promoter (in the stele, cortex and epidermis), *ZIP9* promoter (in the cortex and epidermis) and *IRT3* promoter (in the stele, cortex and epidermis), similarly to previous observations [30]. We could not detect *ZIP12* promoter activity with our construct in our zinc deficiency conditions. Overall, these data suggest a putative additional role of ZIP4, ZIP9 and IRT3 in Zn uptake from the soil at the root periphery under low zinc availability.

Based on our expression analyses and promoter reporter lines characterization, we selected ZIP2, ZIP3, ZIP5 and ZIP8, expressed at the root periphery already in control conditions (*i.e.,* non-deficient), for further investigation as potential candidates for metal uptake from the soil. ZIP2 was of particular interest due to its outer promoter activity, its strong expression in control conditions, and its ability to complement *ctr1* and *smf1* yeast strains. Given these features, we hypothesize that ZIP2 may play a role not in Zn uptake, but in the acquisition of Cu and Mn from the soil (Figs 1B, 1C, S2A and S2C) [18,22]. ZIP8, on the other hand, was investigated as a candidate for Fe acquisition due to its specific epidermal promoter activity, increased expression under Fe deficiency, and its close homology to IRT1 (Figs 1A, 1C, and S2D). ZIP3 and ZIP5 were selected as prime candidates for Zn acquisition from the soil based on multiple lines of evidence. Both genes were expressed at higher levels under Zn deficiency (Figs 1B and S2A) and are known to be regulated by the transcription factors *bZIP19* and *bZIP23* [31]. Additionally, their promoter activity was primarily detected at the root periphery (Figs 1C and S2B), and both have been shown to complement the yeast *zrt1zrt2* strain, further supporting their role in Zn acquisition [18,28]. In addition, both were very recently shown to redundantly act in Zn uptake in an independent study [30].

## ZIP2, ZIP3, ZIP5 and ZIP8 localize to the epidermis-soil interface

To confirm the potential role of ZIP2, ZIP3, ZIP5 and ZIP8 in metal acquisition from the soil, we first examined their localization in roots. Transgenic lines were generated expressing a fusion of the *ZIP* genomic DNA with the coding sequence for mCitrine (mCit), under the control of their endogenous promoters, within their respective loss-of-function mutant backgrounds to ensure functional complementation (see below and S2 Table). Based on previous studies that demonstrated the successful integration of mCitrine in the second extracellular loop of IRT1 while maintaining its functionality [34], we applied the same strategy for ZIP2/3/5/8-mCitrine constructs. Structural modeling of the fusion proteins predicted that inserting mCitrine into these ZIPs transporters did not alter their 3D conformation, similar to IRT1-mCitrine (S3 Fig). Since we expected these transporters to localize at the PM under certain conditions, we co-localized the ZIP transporters with FM4–64, a marker for the PM and endocytic system.

In control, Cu- or Mn-deficient conditions, ZIP2-mCitrine predominantly accumulated in the epidermis of root zone II and in both epidermal and cortical cells of the differentiated root zone III (Fig 2A). Some signal was also detected in the vasculature. ZIP2-mCitrine showed partial PM localization under control conditions but was also present in intracellular structures, which co-localized with FM4–64 after 10 minutes, suggesting its localization in endosomal compartments (S4A and S4B Fig). This dual PM-endosomal localization is common for PM proteins fused to highly sensitive fluorophores, possibly reflecting intermediate trafficking stages from the endoplasmic reticulum (ER) to the PM, as observed for transporters like IRT1, NRAMP1 (NATURAL RESISTANCE-ASSOCIATED MACROPHAGE PROTEIN 1) and PHO1 (PHOSPHATE 1) [35–37]. For IRT1, such localization also reflects substrate-controlled cycling between the PM and early

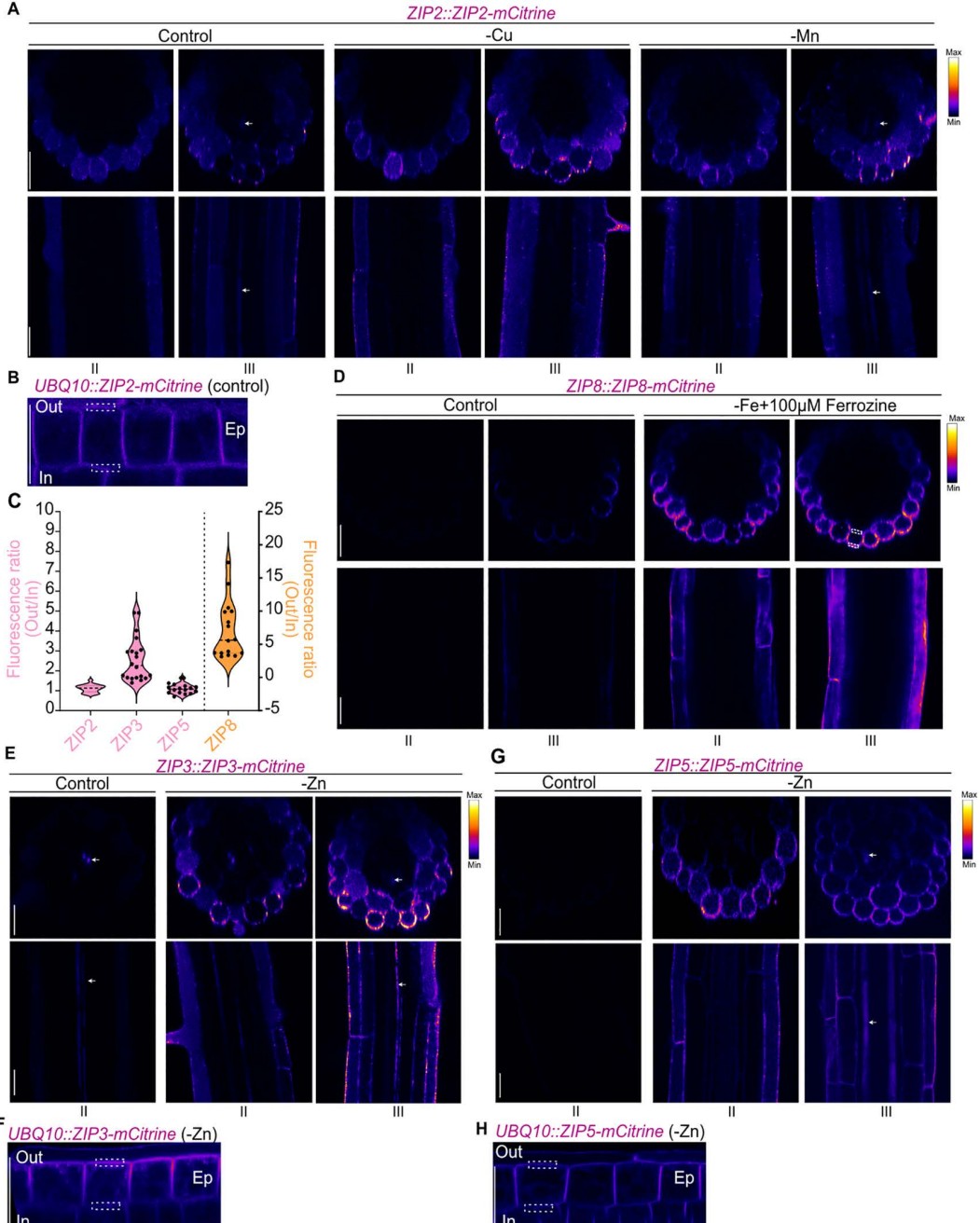

**Fig 2. Localization of ZIP2, ZIP8, ZIP3 and ZIP5 in the outer root domain.** (**A**, **D**, **E** and **G**) Representative confocal images showing mCitrine fluorescence in 5-day-old seedlings expressing ZIP2, ZIP8, ZIP3 or ZIP5 fused to mCitrine under the control of their endogenous promoters. Seedlings were grown in either control or metal-deficient conditions (-Cu, -Mn, -Fe + 100 μM Ferrozine, -Zn). The upper panels display root transversal views (extracted from Z-stacks), while the lower panels show longitudinal views taken at the median focal plane of the root. The corresponding root zone (II or III, as defined in Fig 2C) is indicated below the images. All images were acquired from T2 plants. (**A**, **B**, **D-H**) mCitrine fluorescence is displayed using a Fire look-up-table (LUT). (**B, F, H**) Confocal images of mCitrine fluorescent signal in 5-day-old seedlings expressing *UBQ10::ZIP2/3/5-mCitrine* grown under control (ZIP2) or -Zn conditions (ZIP3 and ZIP5). Roots were imaged in the median focal plane of epidermal (Ep) cells within the elongation zone to assess polar localization. (**C**) Fluorescence ratio analysis for ZIP2, ZIP3, and ZIP5-mCitrine shown in panels **B, F** and **H** (pink). The ZIP8-mCitrine ratio (orange) was calculated in the epidermis of root zone III. The fluorescence ratio was determined by dividing the average fluorescence intensity at the outer (Out) PM by the average intensity at the inner (In) PM for each epidermal cell. The same region of interest (ROI) was used for both outer and inner fluorescence quantification. An example of the ROI used to quantify Out/In ratio is shown with the dashed square for each genotype (**Panels B, D, F, H**).

Ratios were calculated from at least 5 cells per root and at least 3 independent roots. The dashed lines in the violin plots represent the median, while the dotted lines represent the first and third quartiles (n ≥ 15). Scale bars: 25 µm for all images.

endosomes/Trans-Golgi Network (EE/TGN) [34,37,38]. In ZIP2, Cu and Mn deficiency did not notably alter cell-specific localization, except for the near absence of ZIP2-mCitrine in the stele under Cu-deficient conditions (Fig 2A). Considering that transporter polarity can affect their function, we evaluated the potential polarity of ZIP2 in the elongation zone using *UBQ10::ZIP2-mCitrine* overexpression line (Fig 2B). This line was used because ZIP2 expression under its endogenous promoter was undetectable in the elongation zone, where polarity can be more accurately assessed. By calculating the ratio of outer to inner PM fluorescence, we found ZIP2 to be apolar in the epidermis, with a fluorescence ratio close to 1 (Fig 2C).

Next, we investigated ZIP8 localization, given its potential role in Fe acquisition. Under, control conditions, ZIP8-mCitrine was barely detectable in root zone II but was observed at low levels in the epidermis of root zone III (Fig 2D). Under Fe deficiency (-Fe + 100 µM Ferrozine), ZIP8-mCitrine strongly accumulated at the PM of the epidermal cells in both zone II and III (Figs 2D and S4C). Interestingly, ZIP8-mCitrine displayed pronounced outer polar localization in the epidermis under both control and Fe-deficient conditions, reinforcing its possible involvement in metal acquisition from the soil (Figs 2C and S4C).

We also examined the localization of ZIP3 and ZIP5. In control conditions, ZIP3-mCitrine was detected in a few cells in the vasculature (white arrows, Fig 2E). Under Zn-deficient conditions, ZIP3-mCitrine accumulation significantly increased, with prominent expression in the epidermis and cortex, in addition to the vasculature (white arrows). ZIP3-mCitrine predominantly localized to the PM in the elongation zone, though subcellular localization was also observed, co-localizing with FM4–64 indicating these structures were part of the endocytic system (S4D and S4E Fig). Due to the low accumulation of ZIP3-mCitrine in the root tip, we assessed ZIP polarity in the elongation zone using the overexpression line, revealing outer polar localization (facing the soil) in the epidermis under Zn deficiency (Fig 2C and 2F). Lastly, ZIP5-mCitrine localization was explored. In control conditions, ZIP5-mCitrine was undetectable in roots, but it accumulated in the epidermis of root zone II under Zn deficiency (Fig 2G). In zone III, ZIP5-mCitrine accumulated in the epidermis, cortex and some cells in the vasculature under Zn-deficient conditions (white arrows, Fig 2G). ZIP5-mCitrine predominantly localized to the PM but unlike ZIP3, did not show polar localization in the epidermis (Figs S4F, S4G, 2C, and 2H).

## ZIP2 mediates Cu acquisition

Given ZIP2's expression at the root periphery and its PM localization in epidermal cells under Cu and Mn deficiency, as well as its ability to complement *smf1* and *ctr1* yeast strains, we hypothesized that ZIP2 could facilitate Cu and Mn acquisition from the soil [18,22]. To investigate this, we first isolated the previously described T-DNA insertion line *zip2–1*, which exhibits no detectable *ZIP2* expression (S5A and S5B Fig) [18] and generated a new CRISPR/Cas9-induced allele, *zip2–2$_{cr}$* (S5A Fig). The *zip2–2$_{cr}$* mutant harbors a 68 bp deletion resulting in a premature stop codon (S5C Fig). When grown in soil, *zip2–2$_{cr}$* mutants exhibited a slight increase in shoot weight compared to both WT and *zip2–1* (S5D and S5E Fig). However, when plants were grown on Murashige and Skoog-based agar plates containing 0.05 µM Cu (+Cu), shoot FW, root length and chlorophyll content were similar across all genotypes (WT, *zip2–1,* and *zip2–2$_{cr}$*) (Fig 3A–D). In contrast, when Cu was omitted from the growth medium (-Cu), both *zip2–1* and *zip2–2$_{cr}$* exhibited reduced shoot FW and root length compared to WT plants (Fig 3A–D). This increased sensitivity to Cu deficiency in the *zip2* mutants suggests a contribution of ZIP2 to Cu acquisition. To explore whether ZIP2 might also mediate Mn uptake, we grew *zip2* mutants and WT plants on Mn-deficient plates. However, unlike under Cu deficiency, neither *zip2–1* nor *zip2–2$_{cr}$* displayed notable differences compared to WT under Mn-deficient condition (-Mn) (S5F–H Fig). The lack of a pronounced phenotype suggests that ZIP2 plays no or a limited role in Mn uptake. To further confirm ZIP2's role in Cu and Mn acquisition, we

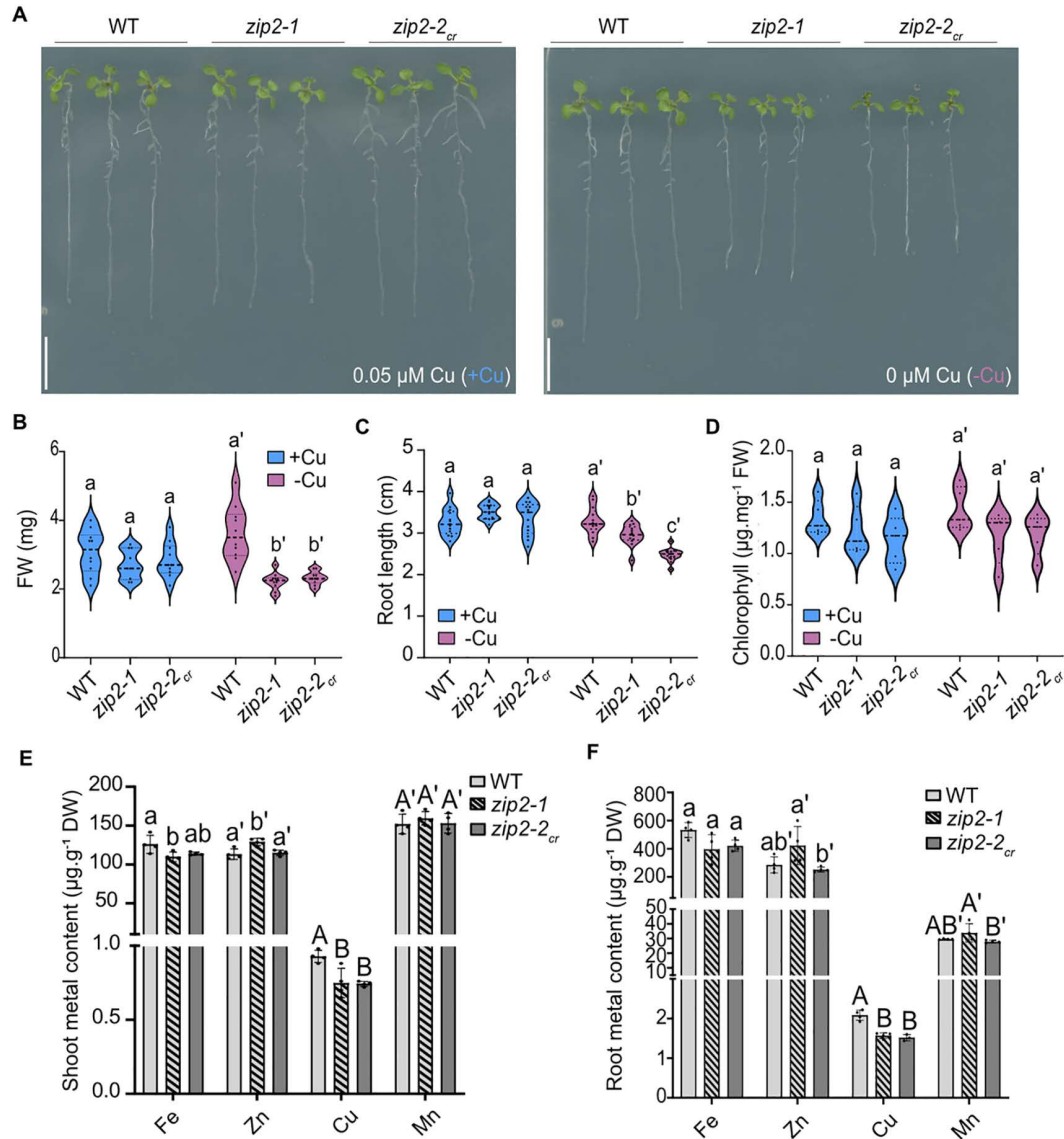

**Fig 3. Impaired Cu accumulation in the *zip2* mutant. (A)** Representative images of WT and two *zip2* mutant alleles grown vertically for 10 days under control conditions (+Cu) and copper deficient conditions (-Cu). Three representative plants for each genotype were transferred to agar plates for imaging. Scale bars: 1 cm. **(B-D)** Violin plots showing the distribution of **(B)** fresh weight (FW), **(C)** root length, and **(D)** chlorophyll content of plants grown in +Cu or -Cu for 10 days (n ≥ 15 for B and C, n ≥ 5 for D). Dashed lines of the violin plots represent the median, while dotted lines mark the first and third quartiles. Different letters indicate statistically significant differences between genotypes, as determined by one-way ANOVA followed by Tukey's post hoc test or Kruskal–Wallis test followed by Dunn's test ($P<0.05$). **(E-F)** Metal content (Fe, Cu, Mn, Zn) in shoots (left) and roots (right) of WT, *zip2–1* and *zip2–2_{cr}* mutant grown vertically for 3 weeks on control agar plates (+Cu). For each biological replicate, 3–4 plants were pooled (n = 4). Data are represented as mean ± SD, with significant differences between genotypes determined one-way ANOVA followed by Tukey post hoc test ($P<0.05$).

measured Fe, Zn, Cu and Mn content in both shoots and roots of *zip2* mutants grown for 3 weeks on control agar plates (+Cu). These metals were selected as they represent the primary substrates for ZIP transporters. Notably, both *zip2–1* and *zip2–2_{cr}* mutants accumulated less Cu than WT in both roots and shoots, reinforcing the idea that ZIP2 plays a role in Cu acquisition (Fig 3E and 3F). Given the reduction in Cu content in *zip2–1*, we used this phenotype to assess the functionality of the ZIP2-mCitrine fusion protein used in localization studies (Fig 2A). Complementation of Cu content in roots

of two independent *zip2–1/ZIP2::ZIP2-mCitrine* lines confirmed the functionality of the fusion protein and the causal role of ZIP2 in this phenotype (S5I Fig). The increased root Mn content observed in the complemented lines suggests that ZIP2 may also facilitate Mn transport into the root (S5I Fig). Overall, our data strongly suggest that ZIP2 primarily mediates Cu acquisition from the soil, with a potential, though more limited, role in Mn acquisition.

## ZIP8 mediates Fe acquisition

Given ZIP8's protein localization and expression patterns (Figs 2D and S2D), we hypothesized that ZIP8 may contribute to Fe uptake from the soil, similarly to its closest homologue, IRT1 (Fig 1A). As little functional data are currently available for ZIP8, we tested this hypothesis using a yeast complementation assay in the *fet3fet4* double mutant, which lacks both high- and low-affinity iron-uptake systems. We expressed *IRT1* cDNA as a positive control, along with two predicted versions of *ZIP8* cDNA (Fig 4A). As expected, IRT1 expression restored growth of the *fet3fet4* mutant even in the presence of 10 µM BPDS, a strong Fe chelator. Notably, ZIP8 expression also partially rescued the growth defect, indicating that ZIP8 can mediate Fe transport in this heterologous system.

To investigate whether ZIP8 can also contribute to Fe transport *in planta*, we generated two CRISPR/Cas9-induced alleles, as no T-DNA insertion line were available for the 5' coding region of *ZIP8*. The *zip8–1$_{cr}$* allele carries a deletion in the promoter and the 5' coding sequence, predicted to produce a truncated 2 aa peptide (S6A and S6B Fig), while *zip8–2$_{cr}$* harbors a single A insertion in the 5' coding sequence, leading to a premature stop codon and a 43 aa protein (S6A and S6C Fig). We began by investigating the phenotype of WT, *zip8–1$_{cr}$*, *zip8–2$_{cr}$*, and *irt1–2* plants grown on soil for 3 weeks. Since IRT1 is the only known ZIP transporter involved in Fe uptake, we used *irt1–2* as a reference mutant. Unlike *irt1–2*, which exhibited strong chlorosis, neither of the *zip8* mutants showed obvious chlorotic symptoms when grown on soil (S6D Fig). However, both *zip8* mutant lines displayed a slight reduction in shoot FW compared to WT (S6E Fig). Next, we grew WT and the two *zip8* mutants for 10 days on control (+Fe, 50 µM Fe-EDTA) and Fe-deficient (-Fe, 0 µM Fe) Murashige and Skoog-based agar media to assess their sensitivity to Fe deficiency. We analyzed shoot FW, root length and chlorophyll content, as indicators of Fe deficiency response. However, no significant difference in these traits were observed between WT and *zip8* mutants under either Fe condition (Fig 4B–E), suggesting that ZIP8 does not play a major role in Fe deficiency. To further explore ZIP8's role in metal acquisition, we measured metal content (Fe, Cu, Mn, and Zn) in shoots and roots of WT and *zip8* mutants grown on +Fe agar plates for 3 weeks. While Mn, Zn and Cu levels were comparable between WT and *zip8* mutants in roots, Fe content in the roots of *zip8* alleles was reduced compared to WT (Fig 4F and 4G). Interestingly, no significant changes in metal content were observed in the shoots of *zip8* mutants, suggesting that ZIP8 primarily mediates metal acquisition in the outer root tissues without substantially affecting root-to-shoot translocation. To confirm the functionality of the ZIP8-mCitrine fusion protein, we used the *UBQ10::ZIP8-mCitrine* reporter line (S2 Table). This line showed increased Zn and Mn content in roots, indicating that the fusion protein used for protein localization was functional (Figs 2D, S4C and S6F). Similarly, overexpression of *IRT1* increased Zn and Mn levels in roots, without significantly altering Fe content (S6G Fig).

In conclusion, our results demonstrate that ZIP8 is involved in Fe acquisition in the outer root tissues and, similar to IRT1, is likely involved in the transport of other divalent ions such as $Cu^{2+}$, $Mn^{2+}$, and $Zn^{2+}$. However, ZIP8's role in Fe transport appears to be more restricted than IRT1.

## ZIP3 and ZIP5 mediate Zn acquisition

Given the localization of ZIP3 and ZIP5 at the root periphery under Zn-deficient conditions (Fig 2E and 2G), along with prior evidence from yeast *zrt1zrt2* complementation assays, we hypothesized that these transporters could be involved in Zn uptake from the soil [18,28]. Due to the close phylogenic relationship between ZIP3 and ZIP5 (Fig 1A), we generated a double mutant (*zip3–1zip5–1*) by crossing the single KO T-DNA insertion mutants *zip3–1* and *zip5–1* to account for potential functional redundancy (S7A–C Fig). Additionally, we created CRISPR/Cas9 mutants for *ZIP3* and *ZIP5* (*zip3–2$_{cr}$*

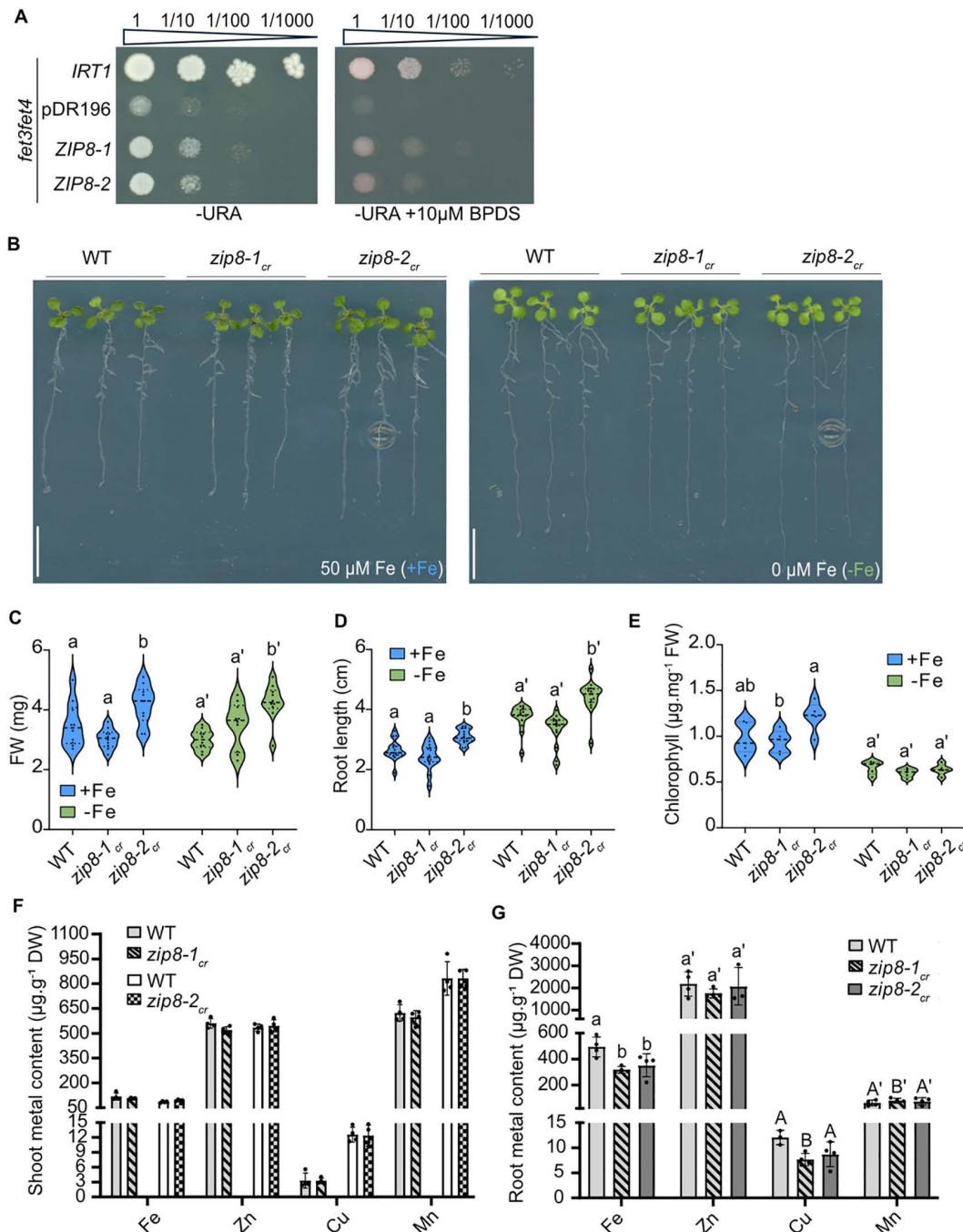

**Fig 4. Impaired Fe accumulation in *zip8* mutant.** **(A)** Growth of the *fet3fet4* mutant transformed with empty vector pDR196 (negative control) or with pDR196 containing *IRT1* (positive control), or two *ZIP8* cDNA version (*ZIP8–1* and *ZIP8–2*). Ten-fold serial dilutions of overnight grown liquid cultures were spotted on -Ura medium, containing or not 10 μM BPDS. **(B)** Representative images of WT and two *zip8* mutant alleles grown vertically for 10 days under control conditions (+Fe) and Fe-deficient conditions (-Fe). Three plants per genotype were transferred onto an agar plate for imaging. Scale bars: 1 cm. **(C-E)** Violin plots showing the distribution of **(C)** FW, **(D)** root length, and **(E)** chlorophyll content of plants grown as described in panel A (n ≥ 15 for B and C, n ≥ 5 for D). Dashed lines of the violin plots represent the median, while dotted lines mark the first and third quartiles. Different letters indicate statistically significant differences between genotypes, as determined by one-way ANOVA followed by Tukey's post hoc test or Kruskal–Wallis test followed by Dunn's test ($P < 0.05$). **(F-G)** Metal content (Fe, Cu, Mn, Zn) in shoots (left) and roots (right) of WT, *zip8–1_cr* and *zip8–2_cr* mutants grown vertically for 3 weeks on control conditions. For each biological replicate, 3–4 plants were pooled (n=4). Data are represented as mean±SD, with significant differences between genotypes determined by Student's *t-t*est or Mann-Whitney test for shoot data (no significant differences observed), while root data

were analyzed using one-way ANOVA followed by Tukey post hoc test and different letters indicate statistically significant differences between genotypes for a given element ($P < 0.05$). For shoots analysis (**F**), data for *zip8-1$_{cr}$* and *zip8-2$_{cr}$* were obtained from 2 independent experiments.

and *zip5–2$_{cr}$*, S7B and S7C Fig). The *zip3–2$_{cr}$* mutant carries a substitution of two CC nucleotides with a single A, resulting in a premature stop codon, while the *zip5–2$_{cr}$* mutant has a single T insertion that also introduces a premature stop codon (S7B and S7C Fig). We first assessed the growth of WT, *zip3–1, zip5–1, zip3–1zip5–1* and the CRISPR-induced mutants *zip3–2$_{cr}$* and *zip5–2$_{cr}$* on soil. No significant differences in FW were observed between WT and the mutants (S7D and S7E Fig). When grown on Murashige and Skoog-based agar plates with sufficient Zn (15 µM Zn, Control), the *zip3–1zip5–1* double mutant exhibited larger overall growth compared to WT and the single mutants, but no significant differences in root length or chlorophyll content were noted (Fig 5A–E). Under Zn-deficient conditions (-Zn), WT plants and the single mutants displayed increased root length, consistent with previous findings [32,39]. Strikingly, the double mutant showed around 40% reduction of shoot FW and chlorophyll content compared to WT under Zn deficiency, indicating heightened sensitivity to low Zn (Fig 5C–5E). Because the reduction in FW and chlorophyll content of the *zip3zip5* double mutant remained modest after 10 days of growth compared to WT plants, and no phenotype was observed in the single mutants, we extended the treatment for an additional 10 days under both +Zn and -Zn conditions. As expected, after prolonged Zn deficiency, the *zip3zip5* double mutant displayed reduced FW and chlorophyll content compared to WT, while the single mutants still showed no detectable phenotype (S7F–H Fig), suggesting that ZIP3 and ZIP5 might function redundantly in Zn acquisition.

To further support the role of ZIP3 and ZIP5 in Zn acquisition, we measured metals content in roots and shoots of 3-week-old plants grown on control Murashige and Skoog-based agar plates containing 15µM Zn. While Fe, Zn, Cu and Mn levels in shoots were similar between WT and *zip3–1zip5–1*, Zn levels in roots were reduced in the double mutant, confirming that ZIP3 and ZIP5 are involved in Zn acquisition (Fig 6A and 6B). Because Murashige and Skoog based medium contains high levels of Zn that could potentially distort metal homeostasis, we also measured shoot metal content in soil-grown plants. Interestingly, we observed lower Zn (and, to a lesser extent, Cu) levels in *zip3zip5* double mutants' shoots compared to WT, whereas Fe and Mn levels remained unaffected, confirming the role of ZIP3 and ZIP5 in Zn acquisition, especially in conditions where Zn concentration is low, and possibly under conditions involving evapotranspiration (Fig 6C). Transpiration and root pressure can indeed impact mineral translocation as shown particularly well for boron [40].

Given the accumulation of ZIP3 and ZIP5 in the vasculature, we wondered if they could contribute to Zn distribution to sink tissues. However, the similar root-to-shoot Zn translocation ratios between WT and *zip3–1zip5–1* suggest that their contribution to Zn partitioning is limited and that other transporters are also likely involved (S8A Fig) [24]. In this respect, the higher mRNA abundance of *IRT1, IRT3, ZIP9* and *ZIP12* in the roots of *zip3–1zip5–1* compared to WT could reflect a role of these transporters in root-to-shoot Zn translocation in the double mutant where they might take over ZIP3 and ZIP5 function in Zn distribution to sink tissues (S8B Fig). This is also supported by the activity of *IRT1* and *IRT3* promoters in the vasculature, which would be compatible with a role in root to shoot translocation (Fig 1C) [27,41].

To confirm the involvement of ZIP3 and ZIP5 in Zn uptake from the soil, we performed short-term zinc uptake assays using roots of WT and *zip3–1zip5–1* double mutants grown under hydroponic conditions. Plants were cultivated for 20 days in Zn-replete hydroponic medium and then transferred to Zn-deficient medium for 1 week to deplete internal Zn stores and induce Zn acquisition systems. Upon Zn resupply, we measured Zn accumulation in roots after 3 and 6 hours. Six hours after resupply, Zn accumulation in *zip3–1zip5–1* mutant was a threefold lower than in WT roots (Fig 6D). This result confirmed that ZIP3 and ZIP5 are required for efficient Zn uptake in roots.

To dissect the individual contributions of ZIP3 and ZIP5, we examined metal content in the single mutants. Since Zn accumulation defects were more visible on soil-grown plants compared to plants grown on Murashige and Skoog-based

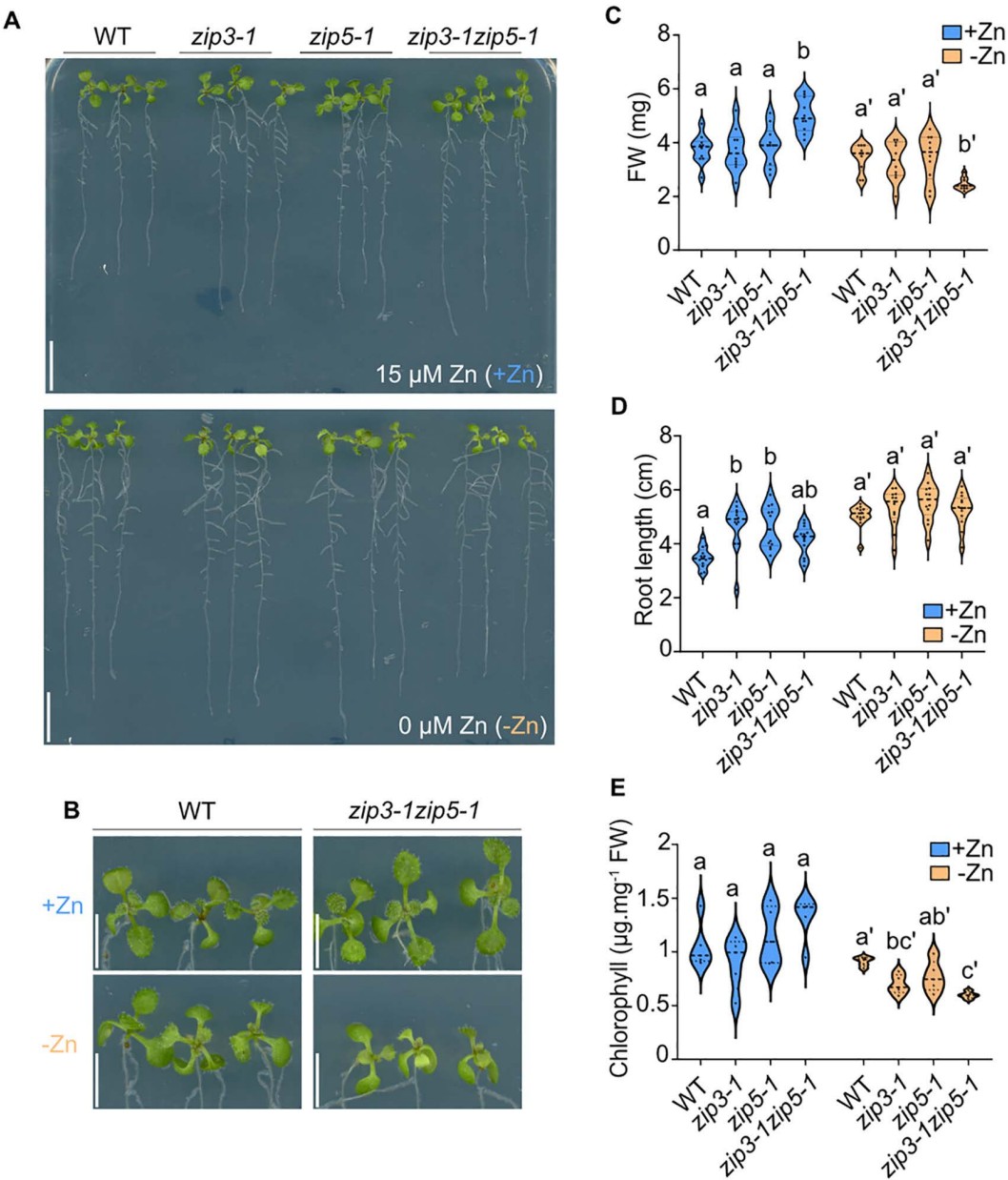

**Fig 5. Phenotype of the *zip3*, *zip5* and *zip3zip5* double mutant(A)** Representative images of WT plants, *zip3-1*, *zip5-1* and the double mutant *zip3-1zip5-1* grown vertically for 10 days under control conditions (+Zn) and Zn-deficient conditions (-Zn). Three representative plants from each genotype were transferred to an agar plate for imaging. Scale bars: 1 cm. **(B)** Zoom of WT and *zip3-1zip5-1* shown in (A). scale bar: 0.5 cm. **(C-E)** Violin plots showing the distribution of (C) FW, **(D)** root length, and **(E)** chlorophyll content of plants grown as described in panel A (n ≥ 15 for C and D, n ≥ 5 for E). Dashed lines of the violin plots represent the median, while dotted lines mark the first and third quartiles. Different letters indicate statistically significant differences between genotypes, as determined by one-way ANOVA followed by Tukey's post hoc test or Kruskal–Wallis test followed by Dunn's test (*P* < 0.05).

agar plates (Fig 6A–C), we analyzed Zn content of *zip3* and *zip5* mutants shoots and roots grown on soil. This analysis revealed that ZIP3 has a primary role over ZIP5 for the Zn deficiency observed in *zip3–1zip5–1*, as *zip5–1* mutants accumulated Zn at levels comparable to WT (Figs 6E and S8C–E). In contrast to *zip3* mutant, no significant differences in root

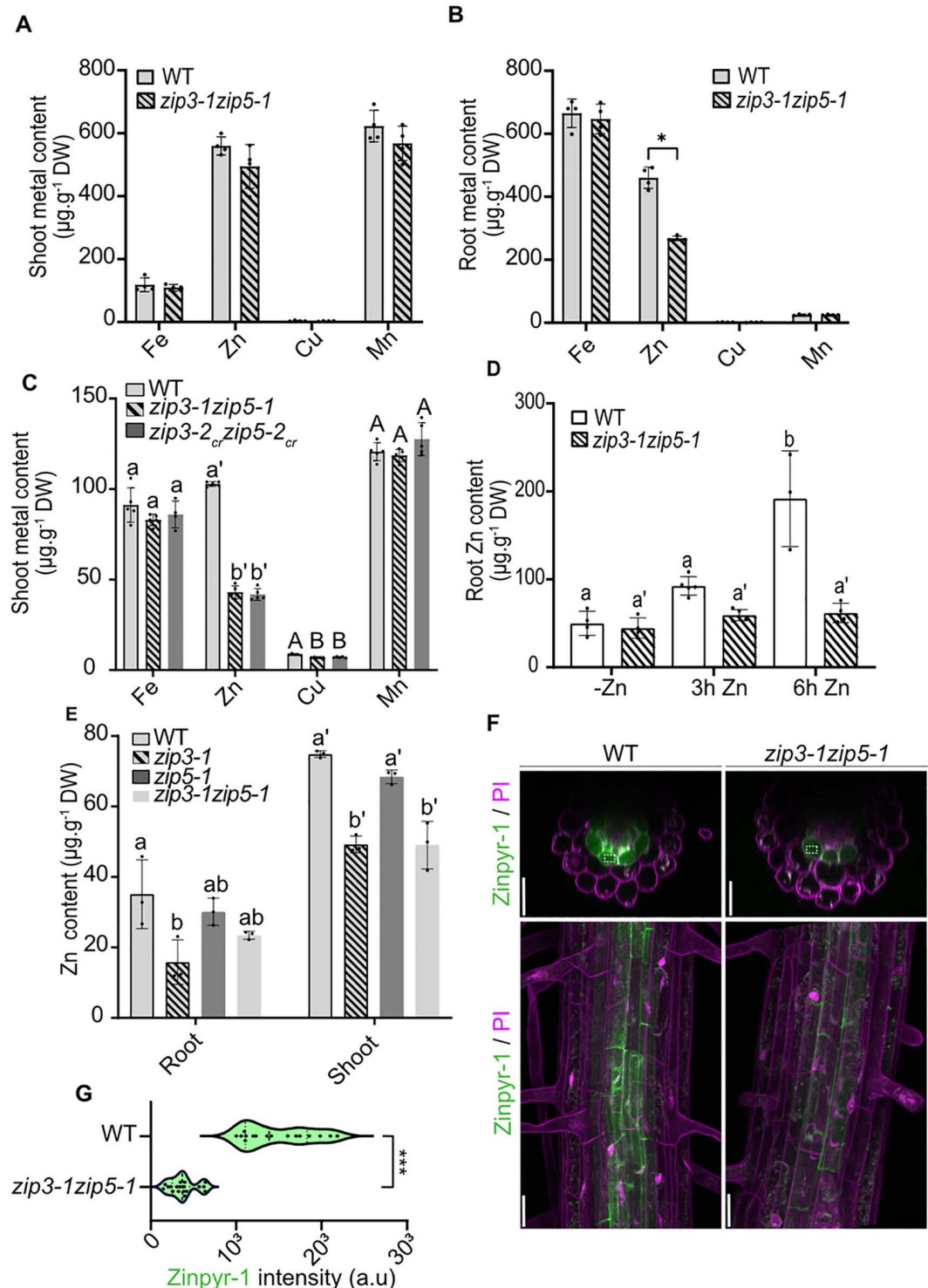

**Fig 6. Ionomic analysis on the *zip3zip5* double mutant.** (A-B) Metal content (Fe, Cu, Mn, Zn) in shoots (left) and roots (right) of WT and *zip3-1zip5-1* mutants grown for 3 weeks on control agar plates. For each biological replicate, 3-4 plants were pooled (n=4). Data are represented as mean±SD, with significant differences between genotypes determined by Student's *t*-test or Mann-Whitney test (*$P<0.05$; **$P<0.01$; ***$P<0.0001$). (C) Metal content (Fe, Cu, Mn, Zn) in shoots of WT, *zip3-1zip5-1 and zip3-2$_{cr}$zip5-2$_{cr}$* mutants grown for 3 weeks on soil. For each biological replicate, 3-4 plants were pooled (n=4). Data are presented as mean±SD, with different letters indicating statistically significant differences between genotypes, determined

by one-way ANOVA followed by Tukey's post hoc test (*P* < 0.05). **(D)** Root Zn uptake in WT and *zip3-1zip5-1* double mutant before (-Zn) and after being resupplied with 10μM Zn for 3h (3h Zn) and 6h (6h Zn). For each biological replicate, 3-4 plant roots were pooled (n = 4). Data are presented as mean ± SD, with different letters indicating statistically significant differences between genotypes, determined by one-way ANOVA followed by Tukey's post hoc test (*P* < 0.05). (E) Zn content in roots and shoots from WT, *zip3-1*, *zip5-1* and *zip3-1zip5-1* grown for 3 weeks on soil. Shoots and roots were harvested from the same plants. For each biological replicate, 3-4 plants were pooled (n = 4). Data are presented as mean ± SD, with different letters indicating statistically significant differences between genotypes, determined by one-way ANOVA followed by Tukey's post hoc test (*P* < 0.05). (F). Representative confocal images of 5-day-old WT and *zip3-1zip5-1* roots labeled with 20 μM Zinpyr-1 for 3h. Plants were grown under Zn-deficient conditions for 5 days prior to staining. Upper panels show cross-sectional views and the lower panels show maximum projections of the differentiated root. Scale bars: 25μm. (G) Violin plots showing the distribution of endodermal Zinpyr-1 signal quantified in region of interest (ROI) in WT and *zip3-1zip5-1* roots. At least five independent roots were analyzed (n ≥ 15). In the violin plots dashed lines represent the median and dotted lines represent the first and third quartiles. Statistical difference between genotypes was determined using Mann-Whitney test (**P* < 0.05; ***P* < 0.01; ****P* < 0.0001).

and shoot Zn content were detected between WT and the *zip5* mutants, both in Murashige and Skoog-based agar plates and soil conditions, suggesting that ZIP5 plays a more limited role in Zn acquisition (Figs 6E and S8D–F). Importantly, elevated Zn levels in the *zip3–1/ZIP3::ZIP3-mCitrine* grown on soil and to a lower extend in *zip5–1/ZIP5::ZIP5-mCitrine* roots grown on plates confirmed the functionality of the fusion proteins (Figs 2E-H, S8G, and S8H).

To gain further insight into the distribution of Zn in the *zip3–1zip5–1* mutant, we used Zinpyr-1 fluorescent probe [11]. In WT differentiated roots, Zinpyr-1 fluorescence was primarily localized in the stele and endodermis, indicating Zn accumulation in these tissues. In contrast, fluorescence intensity was significantly reduced in the *zip3–1zip5–1* double mutant, though the overall pattern of distribution remained unchanged (Fig 6F and 6G).

In addition, metal measurements and Zinpyr-1 staining in *ZIP3* and *ZIP5* overexpression lines provide further evidence to the role of ZIP3 and ZIP5 in metal accumulation (Fig 7A and 7B). Indeed, overexpression of *ZIP3* using the *35S* promoter leads to increased Zn content in roots from plants grown in Murashige and Skoog-based agar plates (+Zn). However, no overaccumulation of other metals typically transported by ZIP family members (e.g., Cu, Mn, Fe) was observed. Moreover, overexpression of both *ZIP3* and *ZIP5* lead to an increased Zn accumulation across root cell layers as visualized after Zinpyr-1 staining (Fig 7B). Overall, these data indicate that ZIP3 and ZIP5 are partially redundant involved in Zn acquisition from the soil in Arabidopsis, with ZIP3 playing a more dominant role.

The fact that *irt1* mutants accumulate WT-like Zn levels in several studies, raises questions about the extent of IRT1's contribution to Zn uptake in soil-grown plants [27,37]. To address this, we employed a promoter-swap approach, expressing *IRT1* under *ZIP2*, *ZIP3* and *ZIP5* promoters (Figs 7C, 7D, S9A, and S9B). Plants expressing *ZIP2::IRT1* and *ZIP3::IRT1* in the *zip3–1zip5–1* background restored Zn accumulation to WT levels in shoots, confirming IRT1's capacity for Zn transport *in planta* (Fig 7D). Overexpression of *IRT1* (*35S::IRT1*) leading to an increase in root Zn levels further corroborated this finding (S6G Fig). However, *ZIP5::IRT1* failed to rescue Zn levels in *zip3–1zip5–1* shoots, likely due to the lack of *ZIP5* promoter activity in the vasculature under our conditions (Fig 2C). Aligning with these data, the root length of *zip3–1zip5–1* mutants expressing *ZIP2::IRT1* and *ZIP3::IRT1* were similar to WT plants in -Zn conditions, contrary to *ZIP5::IRT1* expression in *zip3–1zip5–1* where no complementation was observed.

Overall, these results suggest that Zn acquisition predominantly depends on ZIP3 activity at the root periphery and that ZIP5 and IRT1 can also contribute to Zn acquisition in certain conditions.

### Zn excess triggers ZIP3 endocytosis

Nutrient transporters are frequently regulated at multiple levels, including transcriptional, post transcriptional and post translational mechanisms, in response to nutrient availability [19,34,37,42–44]. To investigate if Zn availability influences ZIP3 stability in the root, we exposed *UBQ10::ZIP3-mCitrine* plants to 150 μM Zn, a 10-fold excess relative to our standard conditions. We first used Propidium Iodide (PI) to control that exposing the plants to 150 μM Zn for 16h do not cause cell death at the root tip, which could interfere with the localization and expression of ZIP3-mCitrine (S10A Fig). Since we wanted to focus on Zn post transcriptional effect on ZIP3, we used the constitutive *UBQ10* promoter, to bypass

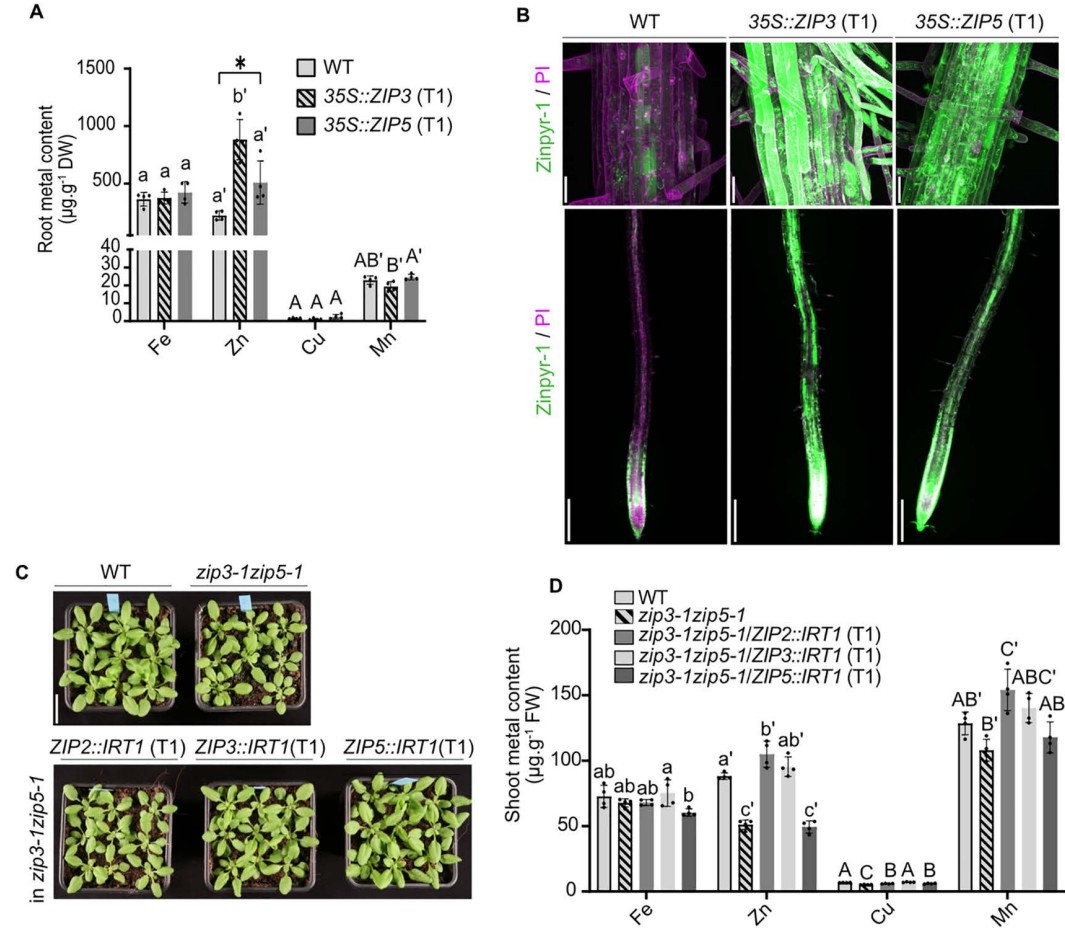

**Fig 7. ZIP3, ZIP5 and IRT1 mediate Zn acquisition.** (A) Metal content (Fe, Cu, Mn, Zn) in roots of WT, ZIP3 and ZIP5 overexpression lines grown for 3 weeks on control agar plates. For each biological replicate, 3-4 T1 plants were pooled (n = 4). Data are presented as mean ± SD, with different letters indicating statistically significant differences between genotypes, determined by one-way ANOVA followed by Tukey's post hoc test (P < 0.05). (B) Representative confocal images of 5-day-old WT, 35S::ZIP3 (T1) and 35S::ZIP5 (T1) plants stained with 20 µM Zinpyr-1 for 3 h. Plants were grown for 5 days on 1 µM Zn prior to staining. Upper panels show a cross section of the differentiated root (scale bars: 25 µm), and lower panels show maximum projections from the root tip (scale bars: 200 µm). (C) Phenotype of WT, zip3-1zip5-1 and zip3-1zip5-1/ZIP2/3/5::IRT1 lines. For each transgenic line, 9 independent T1 were grown per pot. (D) Metal content of the plants grown as described in panel C. For each biological replicate 3-4 independent T1 plants were pooled (n = 4). Different letters indicate statistically significant differences between genotypes for a given element using one-way ANOVA followed by Tukey's post hoc test (P < 0.05).

potential transcriptional regulation of *ZIP3* and focus on post-transcriptional effects. ZIP3 was chosen because of its predominant role in Zn acquisition from the soil. Confocal microscopy revealed that, despite the Zn excess, the overall ZIP3-mCitrine fluorescence did not significantly decrease compared to Zn-deficient condition (Fig 8A and 8B). This suggests that ZIP3 protein abundance is not markedly affected by Zn levels, or that higher Zn concentrations might be required to influence its stability. Although ZIP3-mCitrine accumulation was unchanged, we hypothesized that Zn availability might impact ZIP3 at the post-translational level affecting its subcellular localization, as it is well-established for other transporters that undergo endocytosis in response to excess substrates [34,36,43,45]. Previous colocalization with FM4–64 indicated that ZIP3 is present both at the PM and within the endocytic system (S4E Fig). We examined ZIP3 localization in the root tip, an optimal region for subcellular compartment imaging due to its small cell size. For endosomes

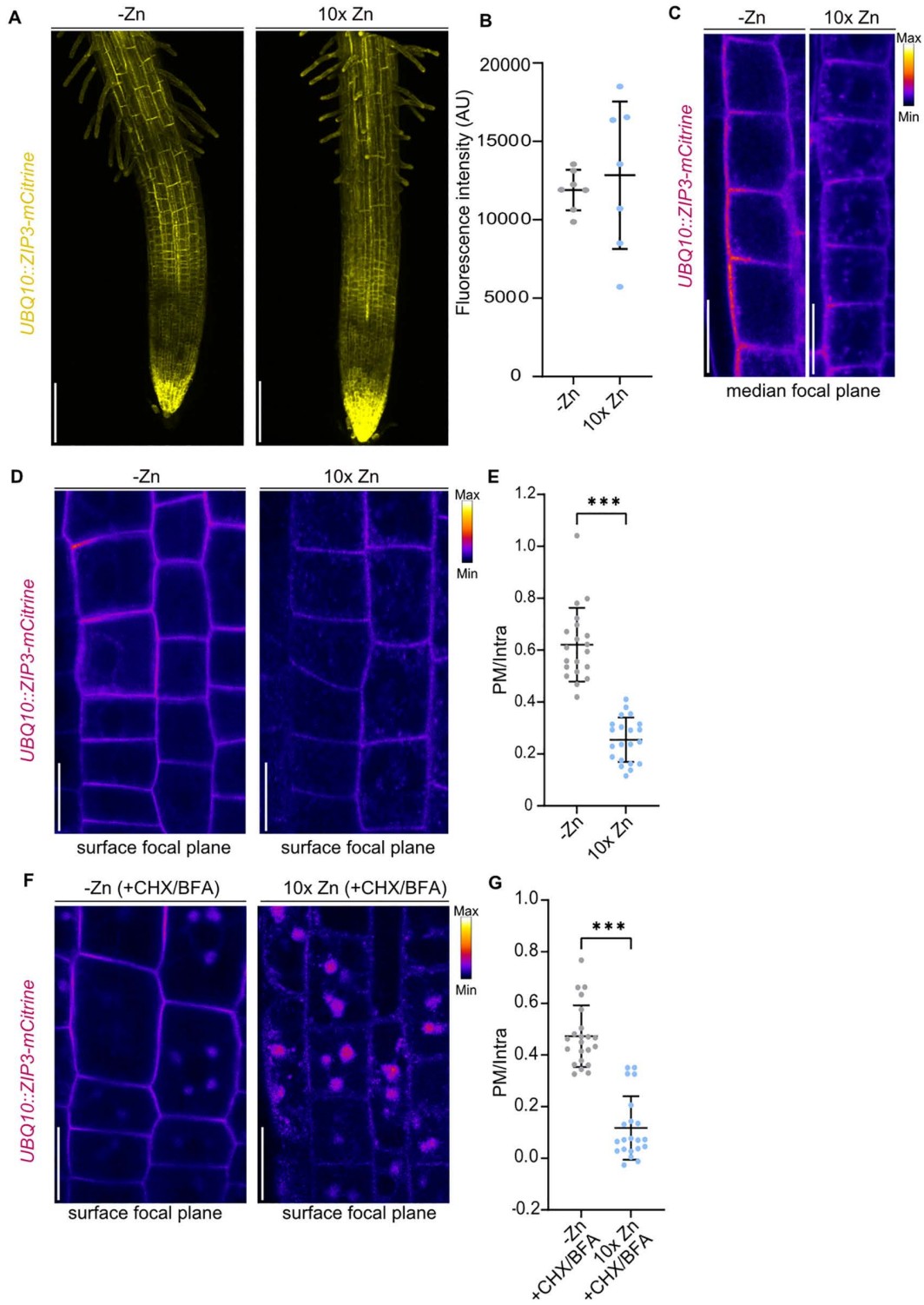

**Fig 8. ZIP3 localization depends on Zn availability.** (A-G) *UBQ10::ZIP3-mCitrine* seedlings were grown on agar plates containing Zn (15 µM) for 5 days, then transferred to liquid medium for 16 h under either Zn-deficient conditions (-Zn) or with a 10-fold excess of Zn (10x Zn; 150 µM) prior to imaging. (A) Representative maximum projections of 5-day-old roots expressing *UBQ10::ZIP3-mCitrine*. The mCitrine signal is shown in yellow. Scale bars: 125 µm. (B) Quantification of ZIP3-mCitrine signal intensity from maximum projection of several independent roots, as shown in panel A (n ≥ 5). Data are represented as mean ±SD. No significant difference between conditions was observed using Mann-Whitney test ($P < 0.05$). (C, D) Close-up views of

the elongation zone in roots shown in panel **A**, with images captured at the median focal plane of the epidermis **(C)** and at the surface of the epidermis (D). The mCitrine signal is displayed using the Fire (LUT) color scale. Scale bars: 12.5 µm. (E) Quantification of the ratio of plasma membrane (PM) to intracellular (intra) of ZIP3-mCitrine signal (n ≥ 20). Data are represented as mean ± SD, and statistical differences between conditions were determined using Mann-Whitney test (*$P < 0.05$; **$P < 0.01$; ***$P < 0.0001$). (F) Surface views of epidermal cells from plants grown as described above and subsequently exposed to 100 µM CHX and 50 µM BFA prior to imaging. The mCitrine signal is displayed using the Fire (LUT) color scale. Scale bars: 25 µm. (G) Quantification of the ratio of PM to intracellular (intra) of ZIP3-mCitrine signal (n ≥ 20) in plants treated as described in panel F. Data are represented as mean ± SD, and statistical differences between conditions were determined using Mann-Whitney test (*$P < 0.05$; **$P < 0.01$; ***$P < 0.0001$).

detection, we imaged the surface focal plane and, for assessing ZIP3 polarity, we focused on the median plane of epidermal cells. Plants were grown in the presence of 15 µM Zn for 5 days, then exposed to either Zn deficiency (0 µM Zn, -Zn) or a 10-fold Zn excess (150 µM Zn, 10xZn) for 16 hours. Under -Zn conditions, ZIP3-mCitrine predominantly localized to the PM, displaying outer polarization, as described earlier (Figs 2C, 2F, and 8C). However, upon exposure to excess Zn, the intracellular ZIP3 signal increased while PM localization diminished (Fig 8C and 8D). Quantification of the PM-to-intracellular (PM/Intra) ratio of ZIP3-mCitrine revealed a significant reduction in this ratio, indicating delocalization of ZIP3 in response to Zn excess (Fig 8E). Notably, we also observed that ZIP3 internalization is triggered at lower Zn concentration, as indicated by a Zn concentration-dependent reduction in the ZIP3 PM/intracellular ratio, with stronger effects observed at higher Zn levels (100 and 150 µM, S10B and S10C Fig).

To determine if this intracellular signal was due to ZIP3 endocytosis upon Zn excess, we treated the plants with Brefeldin A (BFA), a fungal toxin that inhibits vesicle recycling between endosomes and PM. Since BFA also blocks the secretion of newly synthesized PM proteins, we pre-treated plants with cycloheximide (CHX) to inhibit protein synthesis, enabling us to track only the pool of already translated ZIP3-mCitrine and thus focus specifically on post-translational effects. Under -Zn conditions, ZIP3-mCitrine was found at the PM as well as in BFA-induced intracellular bodies, suggesting that ZIP3 cycles between the PM and endocytic compartments under Zn deficiency (8F and S10D Figs). Following Zn excess, the majority of the ZIP3-mCitrine signal accumulates in BFA bodies, as indicated by the reduced PM/Intra ratio (Fig 8F and 8G). This confirms that ZIP3 undergoes rapid endocytosis in response to Zn excess, likely as a protective mechanism to prevent excessive Zn acquisition as previously shown for IRT1 [34,37,38]. Whether ZIP3 is subsequently targeted for vacuolar degradation remains an open question. Interestingly, unlike IRT1 whose localization is regulated by its secondary substrates, ZIP3 localization appears to be modulated at the post translation level by the availability of its primary substrate, Zn.

## Discussion

Understanding the molecular mechanisms controlling metal acquisition by roots is essential for improving plant nutritional quality and growth, particularly under metal-limiting conditions. Despite dicots accounting for one-third of human plant-based nutrition [46], their Zn uptake mechanisms remain understudied compared to monocots. Given the widespread prevalence of Zn deficiency in human populations, enhancing Zn content in crops is critical. This requires identifying key nutrient transporters, understanding their substrate specificity, localization, and regulation, and examining how nutrient deficiencies impact the plant ionome. Here, we investigated the molecular basis of Zn acquisition in Arabidopsis, focusing on the ZIP family of transporters. Using a non-mobile fluorophore, we mapped *ZIP* promoter activities to specific root cell types, uncovering novel roles for 4 ZIP transporters in metal acquisition at the root periphery.

First, we provide reverse genetic evidence that ZIP2 is involved in Cu and potentially also as a Mn acquisition from the soil. While previous studies suggested ZIP2's role in Cu transport, *in planta* evidence was lacking [22]. Our physiological and ionomic analyses confirm ZIP2's role in Cu acquisition at the root periphery, with a possible function in Cu partitioning between roots and shoots. Alongside ZIP2, other Cu transporters such as COPT1 and COPT2 are known to contribute to Cu+ acquisition in Arabidopsis, although their precise localization in root cell types remains unclear [47,48]. The SPL7-(SQUAMOSA PROMOTER BINDING PROTEIN-LIKE7) dependent regulation of *ZIP2*, *COPT1* and *COPT2* underscores

their significance in Cu acquisition, as the transcription factor SPL7 is a key regulator of Cu homeostasis [49]. Our data also support ZIP2's involvement on Mn acquisition as previously suggested [18], expanding the list of known Mn transporters in plants [50]. ZIP2 stands out from other ZIP family members because it is highly expressed and accumulates in the presence of metals, while its expression is repressed under Zn, Mn or Fe deficiencies conditions that typically induce other ZIP transporters (Figs 1B, and S2C, [18,22]).

Second, we demonstrated that ZIP8 is an Fe transporter localizing in a polar manner to the outer epidermis under Fe deficiency. Initially considered a pseudogene [18], *ZIP8* is upregulated in response to Fe deficiency and is part of the FIT- (FER-LIKE FE DEFICIENCY-INDUCED TRANSCRIPTION FACTOR) network, hinting towards a role in Fe homeostasis [51–53]. Its ability to partially complement the yeast *fet3fet4* further supports its role in Fe transport. Reduced Fe accumulation in *zip8* mutant roots confirm ZIP8's role in Fe acquisition. Although the apparent discrepancy between root and shoot metal levels might appear surprising at first, [50] also reported similar discrepancy between root and shoot Mn levels in *nramp1* mutant. It likely reflects that plants may actively prioritize the translocation of essential minerals to the shoot to maintain key physiological functions. The polar localization of ZIP8 to the outer root epidermis reinforces its function in metal acquisition from the soil. Unlike IRT1, ZIP8 does not localize in the stele, suggesting a distinct function in Fe homeostasis. The absence of a macroscopic phenotype in *zip8* mutants and the lack of ZIP8 accumulation in the elongation zone, further indicates that ZIP8 and IRT1 are not functionally redundant.

Third, we provide multi-level evidence that ZIP3 and, to a lesser extent, ZIP5, contribute to Zn acquisition in the outer root domain, similar to OsZIP9 and OsZIP5 in rice. Surprisingly, AtZIP1 and AtZIP12 show higher sequence similarity to OsZIP9 and OsZIP5 than AtZIP3 or AtZIP5 do [15,54]. ZIP3 and ZIP5 are strongly upregulated under Zn deficiency and localize to the PM of the outer root domain. Single and the double mutants phenotypic and ionomic analyses confirm ZIP3's major role in Zn acquisition, while ZIP5 appear to contribute partially redundantly. While our study was in its final phase, independent research reported that ZIP3 and ZIP5 redundantly contribute to Zn uptake in root [30]. If some of our finding rather suggest that ZIP3 and ZIP5 do not fully work redundantly (different protein localization and promoter activity, reduced Zn content in the *zip3* mutant), our findings align with and extend some of their key observations such as promoter activity localization or the biomass reduction of *zip3zip5* double mutant. Some central differences between WT and the *zip3zip5* mutant are also confirmed, such as Zn uptake defect in the *zip3zip5* mutant or the reduced shoot Zn level in the *zip3zip5* mutant. However, several intriguing discrepancies arise. While [30] observed reduced Zn level only in shoots of the double *zip3zip5* mutant, we could also observe reduced Zn level in roots, of both soil and agar grown plants. Similarly, they could not observe reduced Zn accumulation in the single *zip3* or *zip5* mutants, even in conditions allowing evapotranspiration. High Zn level in our MS based media might explain such differences, as metal homeostasis can be altered by elevated Zn concentrations [55]. The concentrations, particularly of Zn and Mn in our plant samples are in this study above plants' requirement. Yet, we confirmed our finding by analyzing soil grown plants. Importantly, our findings also extend the observations of [30] by providing additional evidence of ZIP3 and ZIP5's role with protein localization, overexpressing lines, ionomic data on soil grown plants and Zn effect on ZIP3 internalization.

Unlike IRT1, which broadly impacts metal accumulation when overexpressed [37], ZIP3 and ZIP5 overexpression primarily enhances Zn content, suggesting higher specificity for Zn. This preference was supported by previous studies showing that Zn is the most effective competitor for ZIP3-mediated $^{65}$Zn uptake [26]. In addition, the localization of ZIP3 in the vasculature suggests a role in both Zn acquisition and partitioning. Although we did not observe a reduction in Zn translocation in the *zip3–1zip5–1* double mutant, this may be due to functional redundancy among other ZIPs, as several *ZIP* family members show promoter activity in the stele and are expressed at higher levels in the *zip3zip5* double mutant [18,24,30].

While ZIP3 and to a lesser extent ZIP5 contribute to Zn uptake in roots, it is noteworthy that the double *zip3zip5* mutant still accumulate significant level of zinc in both root and shoots, indicating that other transporters contribute to Zn uptake. The increased expression and promoter activity at the root periphery observed for *IRT3*, *ZIP4* and *ZIP9* under Zn

deficiency suggest that they might also contribute to Zn uptake, under low Zn availability. Zn level quantification in the *zip9* mutant and *IRT3* overexpressing line strongly support this hypothesis [17,56]. The generation of high order mutants would strongly improve our understanding of the contribution of these ZIPs to the overall Zn uptake in roots.

If Zn transporters for acquisition at the outer root domain and for Zn efflux into the xylem have been identified, the radial transport of Zn between these two regions remains unclear. Two plausible, non-exclusive mechanisms could explain this process. First, based on promoter activity analysis in this study and from [30], IRT3, ZIP1, ZIP7, ZIP10 and ZIP11 might facilitate Zn uptake at the endodermal level, as their respective promoters are active in the endodermis. In such scenario, this would suggest that Zn diffuses through the apoplast to reach the endodermis or is exported into the apoplast from cortical cells by an efflux transporter. The presence of the Zn efflux transporter PCR2 in the cortex supports this hypothesis [57]. Zinpyr-1 fluorescence being observed principally in the endodermis and the stele of WT plants support an important role of these tissues for Zn storage. It may explain why only the level of zinpyr-1 fluorescence is altered in the *zip3zip5* mutant but not its distribution across root layers. Alternatively, Zn may be taken up into the cytoplasm at the outer epidermis and transported radially through plasmodesmata via diffusion through their cytosolic sleeves (i.e.,; the space between the desmotubule and the plasma membrane) or via the ER localized transporter MTP2, moving cell-to-cell through plasmodesmata, as already proposed [32]. Since MTP2 is only expressed in Zn-deficient plants, these two scenarios are not mutually exclusive. Although Zn transport through plasmodesmata has yet to be demonstrated, this model would suggest that radial Zn transport does not depend on endodermal Zn transporters or polar efflux Zn transporter across root cell types, as proposed for the coupled transcellular pathway [58]. In addition, the lack of identified polar Zn efflux carriers may indicate that symplastic transport is the preferred route for Zn radial movement.

Additionally, our findings highlight ZIP3 and ZIP8's polar localization at the outer epidermal membrane, suggesting that transporter polarity may be more common in plants than previously recognized [59]. This polarity was shown to be critical for the function of several carriers and is believed to enable directional nutrient flow in roots [58,60,61], as it is well established for PIN- (PIN FORMED-) mediated polar auxin transport [62]. We also observed substrate-induced endocytosis of ZIP3 in response to Zn excess, likely preventing over-accumulation of Zn, similar to IRT1 in Arabidopsis [34]. However, since ZIP3 lacks a histidine-rich stretch in its cytoplasmic domain – a key feature involved in Zn sensing for other transporters – the mechanism by which Zn levels are sensed remains unclear. It is possible that the single histidine between transmembrane domains (TD) 3 and 4 could play a role in this process. Interestingly, ZIP5 also contains a histidine-rich stretch between TD3 and 4, suggesting potential Zn binding and regulation in this region, hinting at similar regulatory mechanisms.

Our study sheds light on the intricate genetic network of metal transport in roots, identifying key transporters and their distinct roles in Zn, Fe, Cu and Mn acquisition. Further research is needed to explore the regulatory mechanisms governing transporter polarity and endocytosis, in order to further optimize nutrient acquisition and homeostasis in plants.

## Materials and methods

### Plant material

All experiments were conducted using *Arabidopsis thaliana* wild type (WT) Columbia-0 background. Mutants and transgenic plants used in this study are as follow: *zip2–1* (SALK_094937, [18]), *zip3–1* (salk_35_B08, [63]), *zip5–1* (SALK_009007C, [29]), all of which were obtained from the Nottingham Arabidopsis Stock Center (NASC). The double mutant *zip3–1zip5–1* was generated by crossing the single mutants *zip3–1* and *zip5–1* and the mutant *irt1–2* was obtained from [64]. Additionally, the following mutants were generated for this study using CRISPR-Cas9 technology: *zip2–2$_{cr}$*, *zip3–2$_{cr}$*, *zip5–2$_{cr}$*, *zip8–1$_{cr}$* and *zip8–2$_{cr}$*. The double mutant *zip3–2$_{cr}$zip5–2$_{cr}$* was generated by crossing the two single *zip3–2$_{cr}$* and *zip5–2$_{cr}$*. Transgenic lines generated for this research include: *pIRT1::NLS-3xmVENUS, pIRT2::NLS-3xmVENUS, pIRT3::NLS-3xmVENUS, pZIP1::NLS-3xmVENUS, pZIP2::NLS-3xmVENUS, pZIP3::NLS-3xmVENUS,*

*pZIP4::NLS-3xmVENUS, pZIP5::NLS-3xmVENUS, pZIP6::NLS-3xmVENUS, pZIP7::NLS-3xmVENUS, pZIP8:: NLS-3xmVENUS, pZIP9::NLS-3xmVENUS, pZIP10::NLS-3xmVENUS, pZIP11::NLS-3xmVENUS, pZIP12::NLS-3xmVENUS, pZIP2::ZIP2-mCitrine, pZIP3::ZIP3-mCitrine, pZIP5::ZIP5-mCitrine, pZIP8::ZIP8-mCitrine, UBQ10::ZIP2-mCitrine, UBQ10::ZIP3-mCitrine, UBQ10::ZIP5-mCitrine, UBQ10::ZIP8-mCitrine, pZIP2::IRT1, pZIP3::IRT1, pZIP5::IRT1, p35S::ZIP3, p35S::ZIP5 and p35S::IRT1.* The construct *pIRT1* was ordered from NASC (N2106308).

## Constructs

Plasmids were generated using Gibson and Multisite Gateway cloning techniques (Thermo Fisher Scientific). A list of primers used for cloning is provided in S3 Table. All DNA were amplified using Phusion Plus DNA Polymerase (ThermoFischer) following manufacturer instructions. *ZIP* promoter sequences upstream of the ATG start codon were amplified from Arabidopsis Col-0 genomic DNA and inserted into a modified *pDONRP4-P1R* (Thermo Fisher Scientific) via Gibson assembly. Briefly, the *ccdB* and the *CmR* cassettes between the AttP1R and attP4 recombination sites were removed in *pDONRP4-P1R* (Thermo Fisher Scientific) and replaced by EcoRV, BglII, XbaI, BamhI cloning sites (S11A Fig). For the generation of promoter-reporter constructs (*promoter::NLS-3xmVenus*), entry plasmids containing the promoter regions were recombined with *pDONRL1-NLS-3xmVenus-L2* and *pEN-R2-tHSP18.2-L3* into the destination vector *pFR7m34GW* [65]. For yeast complementation assays, two *ZIP8* cDNA versions (*ZIP8–1: 888 bp* and *ZIP8–2: 897 bp*) predicted from Aramemon (ARAMEMNON, plant membrane protein database release 8.1) were synthetized by Genewiz. *IRT1* cDNA was cloned using specific primers. *IRT1* and *ZIP8* cDNA were then inserted by Gibson in the pDR196 yeast expression vector [66]. To study endogenous protein localization, *ZIP* genomic sequences were amplified from Col-0 genomic DNA and inserted by Gibson in a modified *pDONR221,* containing cloning sites instead of *ccdB* and *CmR* cassettes (S11B Fig). The mCitrine fluorescent protein was inserted into the second extracellular loop of each ZIP transporter using Gibson assembly [34]. For ZIP2-mCitrine, mCitrine was inserted 153 bp downstream of the ATG start codon; for ZIP3-mCitrine, it was inserted 95 bp from ATG; for ZIP5-mCitrine, 89 bp from ATG; and for ZIP8-mCitrine, 87 bp from ATG. The final destination vectors for expression in plants were obtained using multisite Gateway recombination system (Life Technologies), employing the *pFR7m34GW* destination vector along with the various *pDONR221-ZIP-mCitrine* entry clones to generate the *ZIP::ZIP-mCitrine* constructs. For promoter swap analysis, entry vector containing the promoters of *ZIP2, ZIP3* and *ZIP5* were recombined with *pDONRL1-gIRT1-L2* and *pEN-R2-tHSP18.2-L3* into the destination vector *pFR7m34GW.* Vectors carrying *ZIP2/3/5::IRT1* were transformed in *zip3–1zip5–1* background to evaluate IRT1 function in Zn acquisition. For CRISPR/Cas9 mutant generation, single guide RNA (sgRNA) for Cas9 were designed using webtool Benchling (https://www.benchling.com). Pairs of annealed oligonucleotides of the sgRNA were cloned into the BbsI linearized entry vector recombined into the destination vector containing Cas9 expression cassette controlled by the *PcUBi4-2* promoter and a FastRed selection marker [67]. To generate the *zip2–2$_{cr}$* mutant, a single sgRNA targeting the 5' genomic region was combined with the *ZCas9i* controlled by the *pEC1.2* promoter [68]. For the generation of the *zip3–2$_{cr}$, zip5–2$_{cr}$, zip8–1$_{cr}$* and *zip8–2$_{cr}$*, triple sgRNA targeting the 5' genomic region were employed with *Cas9*. All constructs were sequenced (Sanger DNA sequencing) in pDONR and in the destination plasmid before transformation into Agrobacterium strain GV3101 by electroporation and were subsequently used for Arabidopsis plant transformation via the floral-dip method. T1 plants were first selected based on the red fluorescence of transformed seeds and fluorescent mono-insertional T2 were subsequently selected. All experiments were performed on mono-insertional T2 after red fluorescence sorting unless otherwise specified in figure legends. CRISPR-induced mutation experiments were performed on T3 or T4 homozygous lines.

## Growth conditions

For *in vitro* culture, seeds were surface sterilized in 70% ethanol for 5 minutes with agitation, followed by 3 rinses with absolute ethanol before drying. The sterilized seeds were then sown in lines (density of 12–20 seeds depending of the

experiment) at 1.5 cm from the upper size of 12x12 cm square plates containing 50 ml half-strength Murashige and Skoog (MS) based media with 0.8% agar (Duchefa) and no sucrose (pH 5.7, adjusted with KOH). For metal deficiency or sufficiency, plants were grown in the presence (+Zn, +Cu, +Fe +Mn) or absence (-Zn, -Cu,-Fe, -Mn) of 15 µM Zn, 0.05 µM Cu, 50 µM Fe and 50 µM Mn provided as $ZnSO_4$, $CuSO_4$, Fe-EDTA and $MnSO_4$ respectively. For –metal conditions, the respective metal was not added to the medium, but trace amounts are likely present in the agar. Additional nutrients were supplied at following concentration: 9.4 mM $KNO_3$, 10.3 mM $NH_4NO_3$, 625 mM $KH_2PO_4$, 1.49 mM $CaCl_2$, 0.75 mM $MgSO_4$, 50 µM $H_3BO_3$, 0.5 µM $Na_2MoO_4$, 2.5 µM KI, 0.05 µM $CoCl_2$. After stratification in the dark for 3 days at 4°C, the plates were transferred to growth chamber under continuous day conditions (light intensity ~100 µE) at 22 °C with ~50% humidity and the plants grown vertically. For physiological analysis, root length, FW and chlorophyll contents were assessed in 10-day-old plants grown on control or metal-deficient agar plates. For elemental analysis, 12 plants per plate or 6 plants per pot were grown either on agar control media for three weeks or in soil for the same duration before harvesting. For soil grown plants, plants were harvested at Zeitgeber time (ZT 8) (8 h after lights on). For mRNA relative quantification, twenty 7-day-old seedlings per plate were directly sown and vertically grown on control or metal deficient plates before harvesting. Live-microscopy analyses were conducted on 5-day-old seedlings. For promoter activity, expression analysis and protein localization, plants were directly sown and grown for 5 days on control or metal-deficient agar plate. For short-term metal deficiency or excess treatments, 5-day-old plants were transferred to liquid medium without Zn (-Zn) or with an excess of Zn (10xZn; 150 µM of Zn provided as $ZnSO_4$) for 16 hours before imaging. CHX was applied to plants at a final concentration of 100 µM for 1 hour before treatment with 50 µM BFA, which was maintained for the duration of the BFA treatments (3 hours). For amplification and experiments in soil, plants were grown in long-day conditions (16 hours of light, 8 hours of darkness) with a light intensity of 150–180 µE at 60–70% humidity, and at 20±2 °C.

### Short time Zinc uptake assay

For short time Zn uptake assay, plants were hydroponically grown in a controlled growth chamber (22°C, 70% relative humidity, ~100 µE light intensity) under long day conditions (16 hours of light, 8 hours of darkness). A density of one plant per 80 ml of MS/2 solution was used (same composition and pH as described for *in vitro* conditions). Following 3 weeks of growth, in presence of Zn, plants were transferred to a MS/2 solution without Zn and grown for 1 week in this Zn-deficient condition. This Zn deprived media was renewed once after 4 days. After 1 week of Zn-deficiency, plants were incubated in a MS/2 solution containing 10 µM $ZnSO_4$ for 3 and 6 hours, before roots were harvested. Roots were washed and analyzed as described for the ionomic analysis.

### Confocal Microscopy and image analysis

Cell walls were stained with a 10 µg.ml⁻¹ solution of propidium iodide (PI; Sigma-Aldrich, Cat# P4170) for 10 minutes, while the PM and endocytic compartments were stained with a 4 µM solution of the styryl dye FM4–64 (ThermoFischer scientific, Cat# T13320) for 10 minutes. Zn visualization was conducted with 10 µM Zinpyr-1 (Abcam, Cat# 145349) staining as described by [11]. To prevent excessive Zn accumulation in the roots and minimize fluorescent signal saturation, plants were grown under control conditions with 1 µM Zinc (+Zn) instead of 15 µM. Confocal laser scanning experiments were conducted using a Zeiss LSM 780 system. The excitation (ex) and detection (em) settings were configured as follows: mVenus/mCitrine ex: 514 nm, em: 515–570 nm; propidium iodide (PI) ex: 514 nm, em: 586–679 nm; Zinpyr-1: ex: 488 nm, em: 495–535 nm; and FM4–64 ex: 514 nm, em: 650–742 nm. Images were captured using a 40X objective for Z-stack imaging and a 20X dry objective for longitudinal views. All images were processed using the Fiji software (LUT, orthogonal views, maximum projections, merge, fluorescence quantification, etc) [69]. For Z-stack images of promoter-reporter lines, protein localization and Zinpyr-1 staining, the Z-stack images were resliced vertically to generate transversal views. For promoter reporter lines only, a maximum projection of the mVenus marker channel was then merged

(3D; X, Y, Z) with a representative single stack of the PI-stained cell wall channel (2D; X, Y). For T1 lines, at least five independent lines were observed (selected based on red seed fluorescence) and for T2 lines, at least two independent mono-insertional lines were observed. In all case a representative line is shown in the manuscript. For Fig 8, PM/intra signal were calculated as described in [70]. The fluorescence intensity of Zinpyr-1 in the endodermis was determined using identical regions of interest (ROI). The Zinpyr-1 signal in three to four endodermal cells from at least five independent roots was measured to obtain the average Zinpyr-1 fluorescence intensity.

## qRT-PCR

For gene expression analysis, seedlings were sown (20 per plate) and vertically grown for 7 days. Roots from approximately forty seedlings were harvested and pooled to form one biological replicate. RNAs were extracted using a TRIzol-adapted RNeasy MinElute CleanupKit (Qiagen). Subsequently, RNAs were reverse transcribed with a Thermo Scientific Maxima First Strand cDNA Synthesis Kit following the manufacturer's protocol. Quantitative Real-time PCR was conducted on an Applied Biosystems QuantStudio5 thermo-cycler using Applied Biosystems SYBR Green master mix. The relative expression of *ZIP* genes was quantified with the $2^{-\Delta\Delta Ct}$ method with *Clathrin* as a reference gene (AT5G46630). A list of primers used for qRT-PCR is provided in S3 Table, the primer efficiency is provided in S4 Table and Clathrin expression in S5 Table. Each experiment included 3 technical and at least three biological replicates.

## Ionomic analysis

Three weeks-old seedlings (roots and shoots) were used to quantify metal content. For shoots and roots from Murashige and Skoog-based agar plates and shoot from soil conditions, samples were harvested separately and desorbed by washing in 10 mM Na$_2$EDTA (pH 5.7) for 10 minutes, followed by 3 rinses with ultrapure Milli-Q water for 1 minute each. For soil grown roots analysis, 3 weeks old plants were roughly washed with distilled water to remove most of the soil attached to the roots. Roots were then carefully washed with MilliQ water. Roots were finally immersed in 4°C MilliQ water and soil particles were removed from roots using tweezers until fully cleaned. Once cleaned, roots were desorbed as described above. After desorbing, the tissues were blotted dry with paper towels and placed at 65°C overnight. Once dried, the tissues were ground, and 5–10 mg of dry tissue per biological replicate was mixed with 750 µl of nitric acid (65% [v/v]) and 250 µl of hydrogen peroxide (30% [v/v]). The samples were left at room temperature overnight and then mineralized at 85°C for 24 hours. Following mineralization, 4 ml of Milli-Q water was added to each sample. Four to five biological replicates were analyzed for each genotype. Elemental analysis was performed using inductively coupled plasma optical emission spectroscopy (ICP-OES 5800; Agilent Technologies) using calibration standard solutions provided by the manufacturer. The ICP-OES is calibrated in wavelength each time the machine is switched on, using a 5 ppm calibration solution supplied by the manufacturer (ICP-OES&MP-AES wave cal 6610030100, Agilent Technologies) and performance tests are proceeded (optics, sensitivity and resolution). Quality control was performed with Certified Reference Material strawberry leaf powder (LGC7162), corresponding perfectly to the type of matrices usually handled. Atomic emission wavelengths are selected using the precision obtained from CRM measurements within +/- 10% of the theoretical value. The following elements quantified with wavelength: calcium (Ca 393.366nm), copper (Cu 324.754nm), iron (Fe 238.204nm), potassium (K 766.491nm), magnesium (Mg 280.270nm), manganese (Mn 257.610nm), molybdenum (Mo 202.032nm), sodium (Na 589.592nm), strontium (Sr 407.771 nm) and zinc (Zn 202.548nm).

## Yeast complementation assay

For yeast complementation, the *fet3fet4* double mutant was used [71].The *fet3fet4* strain was transformed using one-step transformation protocol [72], either with the empty pDR196 vector, or pDR196 vector containing *IRT1* or *ZIP8* cDNAs. Positive clones were recovered on -URA (Uracil) selection medium containing 0.078% CSM-URA (Complete Supplement

Mixture), 2% glucose, 0.67% YNB (yeast nitrogen base without amino acids) and 2% agar, and verified by PCR. For drop tests, yeast clones were incubated overnight at 28°C in an agitating incubator in 2ml of -URA liquid media containing 50 μM FeCl$_3$. Cells were then collected, washed twice with sterile MilliQ water, the OD measured and adjusted to the same for all constructs before realizing serial dilutions. For each dilution, 5 μl drops were deposited on the same -URA selection media with or without 10μM BPDS (bathophenanthroline disulfonic acid disodium salt). Yeasts were incubated 5 days at 28°C and pictures taken with an Epson perfection V800 scanner.

## Chlorophyll quantification and F$_v$/F$_m$ measurements

Chlorophylls were extracted from 3 weeks (only for Fig 1C) or 10-day-old plants tissues by incubating sample overnight at 4°C in 10 volumes of dimethylformamide (DMF) (v/w). Following extraction, samples were centrifugated, and chlorophyll content was quantified using a NanoDrop spectrophotometer at wavelength 647 nm and 664 nm. Chlorophyll concentrations were calculated according to the methods previously described by [73]. For Fv/Fm measurement, plants were dark-adapted for 15 minutes before measurements were taken with a FluorCam 800 MF, as described in [74]. The maximum quantum efficiency of photosystem II (F$_v$/F$_m$) was measured by applying saturating pulses of 2000 μmol m$^{-2}$ s$^{-1}$ (white light, 400–720 nm) for 960 ms, followed by detection using orange-red light pulses (620 nm, 10 μs). All measurements were performed at room temperature.

## Phylogenetic tree

Protein sequences for *Arabidopsis thaliana* ZIP family members were retrieved from TAIR (https://www.arabidopsis.org) or Aramemnon (https://aramemnon.botanik.uni-koeln.de/). Phylogenetic relationships were defined using Phylogeny.fr (http://www.phylogeny.fr/).

## AlphaFold and AlphaFold2 protein interaction prediction

Wild type ZIP and ZIP-mCitrine models were predicted using AlphaFold (a) (https://alphafold.ebi.ac.uk/) or the ColabFold v1.5.5: AlphaFold2 using MMseqs2 (b) [75–77]. The number of recycles were set to 3 for ColabFold predictions.

## Statistical analysis

All statistical analyses were conducted using GraphPad Prism software or within the R environment (R Core Team 2023). Data were first assessed for normality and homoscedasticity using the Shapiro-Wilk test and Brown-Forsythe test, respectively. For data meeting these assumptions, pairwise comparisons were performed using a two-tailed Student's *t*-test, while multiple comparisons were conducted using a one-way ANOVA followed by a Tukey's post hoc test. When the assumptions of normality or homoscedasticity were violated, non-parametric tests were employed. In such cases, pairwise comparisons were carried out using the two-tailed Mann-Whitney U test, and multiple comparisons were analyzed using the Kruskal–Wallis test followed by Dunn's post hoc test or a non-parametric version of Tukey's test (R). A significance threshold of $P < 0.05$ was applied in all tests. All experiments were repeated at least 3 independent times except for experiments in S5B and S7A Figs and representative experiments are presented in the manuscript. In all the manuscript, we used the term biological replicates to refer to independently collected pools of plants (ranging from a minimum of 3–4 plants to up to 40 seedlings, depending on the experiment, as specified in the figure legends). These pools were harvested from the same experiment, in which plants were grown and samples analyzed in parallel.

## Accession numbers

The sequence data used in this article (for generating constructs and for qRT-PCR) can be found in the GenBank/EMBL databases under the following numbers: *IRT1* (*AT4G19690*), *IRT2* (*AT4G19680*), *IRT3* (*AT1G60960*), *ZIP1* (*AT3G12750),*

*ZIP2 (AT5G59520)*, *ZIP3 (AT2G32270)*, *ZIP4 (AT1G10970)*, *ZIP5 (AT1G05300)*, *ZIP6 (AT2G30080)*, *ZIP7 (AT2G04032)*, *ZIP8 (AT5G45105)*, *ZIP9 (AT4G33020)*, *ZIP10 (AT1G31260)*, *ZIP11 (AT1G55910)*, *ZIP12 (AT5G62160)*.

## Supporting information

**S1 Fig. Impact of Zinc deficiency on plants development and ionome. (A**) Representative images of WT plants grown for 3 weeks in Zn-sufficient (+Zn) and Zn-deficient conditions (-Zn), displayed on horizontal (top) and vertical plates (bottom). Scale bars: 1 cm. Violin plots showing **(B)** fresh weight (FW), **(C)** chlorophyll content, and **(D)** Fv/Fm of dark-adapted WT plants grown 3 weeks in Zn-sufficient (+Zn, blue) or Zn-deficient conditions (-Zn, orange) conditions. In the violin plots dashed lines represent the median and dotted lines represent the first and third quartiles (n ≥ 10) **(E)** Shoots and **(F)** roots ionome profiles of WT plants grown for 3 weeks under Zn-sufficient (+Zn) or Zn-deficient (-Zn) conditions on horizontal plates as in upper panel A. Data are represented relative to +Zn conditions, set at 100, and correspond to the average values from 3 independent pools, each consisting of at least 10 individual shoots and roots. Numerical values are presented in S1 Table. **B-F** Statistical differences were determined using Student's *t*-test or Mann Whitney test, (*$P < 0.05$; **$P < 0.01$; ***$P < 0.0001$). **(G)** Shoot-to-root ratios of the average elemental content in WT plants grown for 3 weeks on agar plate in presence (+Zn) or absence (-Zn) of zinc. Values correspond to the ratios from 3 independent pools of at least 10 individual shoots and roots (See S1 Table for numerical values).
(TIF)

**S2 Fig. Zinc deficiency affects *ZIP* gene expression. (A)** Relative transcript levels of the 15 *ZIP* members in WT roots from plants grown for 1 week under +Zn or -Zn conditions. Each dot represent one biological replicate. Data are represented as mean ± SD (n = 4, each sample is a pool of at least 40 seedlings). Statistical differences were determined using Student's *t*-test or Mann-Whitney test (*$P < 0.05$; **$P < 0.01$; ***$P < 0.0001$). **(B)** Representative images of *ZIP::NLS-3xmVenus* expression pattern in differentiated root (zone III). Nuclear-localized mVenus signal is shown in green while PI used to stain the cell wall is shown gray. mVenus signal is presented as a maximum projection of Z-stacks (XYZ), overlaid with a single orthogonal view of PI extracted from the Z-stacks. Images correspond to T2 plants; 2 independent T2 lines were imaged. Plants were grown in zinc deficient (-Zn) conditions for 5 days before imaging. Scale bars: 25 µm for all images. **(C-D)** Relative transcript levels of **(C)** *ZIP2* and **(D)** *ZIP8* in WT roots from plants grown under metal-sufficient (Control, CT) or metal-deficient (-Fe, -Cu, -Mn) conditions. Data are represented as mean ± SD (n = 3, each sample is a pool of at least 40 roots). Different letters indicate statistically significant differences between conditions determined by one way ANOVA followed by Tukey post hoc test or Kruskal–Wallis test followed by non-parametric Tukey test ($P < 0.05$).
(TIF)

**S3 Fig. AlphaFold models of ZIP and ZIP-mCitrine.** Predicted 3D structures of WT, IRT1, ZIP2, ZIP3, ZIP5, ZIP8 and their fluorescent versions were generated using AlphaFold (a) or AlphaFold2 (b) respectively. For ZIP5 and ZIP8, we used *ZIP5.1* and *ZIP8.1* transcript versions for structure prediction. TMH: Transmembrane helix.
(TIF)

**S4 Fig. Co-localization of ZIP2, ZIP3, ZIP5 and ZIP8-mCitrine with FM4–64. (A, C, D, F)** Co-localization of ZIP2, ZIP3, ZIP5 and ZIP8-mcitrine with FM4–64 in the epidermis of the root elongation zone (A, D, F) or the differentiated root (C), imaged 10 minutes after FM4–64 staining (top panels). The fluorescence signals of mCitrine and FM4–64 are plotted along the dashed lines in each image to show overlap (bottom panels). Scale bars: 25 µm. **(B, E, G)** Co-localization of ZIP2, ZIP3 and ZIP5-mCitrine with FM4–64 at endosomes in the epidermis of the elongation zone, 10 minutes after FM4–64 staining (top panels). The fluorescence intensity profiles of mCitrine and FM4–64 are plotted across the dashed lines in each image (bottom panels). Scale bars: 6.25 µm. a.u; arbitrary units.
(TIF)

**S5 Fig. Characterization of *ZIP2* mutants and complementation lines. (A)** Schematic representation of the genomic structure of *AtZIP2,* showing the locations of the T-DNA insertion in *zip2–1* and the CRISPR-induced mutation in *zip2–2_cr*. Black boxes represent exons, the black lines represent introns, the dashed line indicates the promoter region, and the black angled arrow marks the START codon. **(B)** Relative *ZIP2* transcript levels in WT and *zip2–1* plants grown for 1 week on control agar plates is shown (n = 4, each biological replicate consisting of a pool of at least 40 roots). Data are represented as mean ± SD, and statistical significance was determined using the Mann-Whitney test (*$P < 0.05$; **$P < 0.01$; ***$P < 0.0001$). **(C)** Partial genomic (top) and protein (bottom) sequence of *ZIP2* in WT and the *zip2–2_cr* mutant. The *zip2–2_cr* allele results in a truncated protein due to a premature stop codon, producing only 50 amino acids (aa) compared to the 353 aa in the WT. **(D)** Phenotype of WT, *zip2–1* and *zip2–2_cr* plants grown for 3 weeks in soil under long-day conditions. Scale bar: 2 cm. **(E)** Quantification of fresh weight (FW) for plants grown as in panel C (n ≥ 10). In the violin plots dashed lines represent the median, and dotted lines represent the first and third quartiles. Different letters indicate statistically significant differences between different genotypes after one-way ANOVA followed by Tukey's post hoc test ($P < 0.05$). **(F-H)** Violin plots representing **(F)** FW (n ≥ 15), **(G)** root length (n ≥ 15) and **(H)** chlorophyll (n ≥ 5) content of WT, *zip2–1* and *zip2–2_cr* plants grown for 10 days under control (+Mn) and manganese-deficient (-Mn) conditions. In all panels, dashed lines represent the median, and dotted lines mark the first and third quartiles. Different letters indicate statistically significant differences between different genotypes using one-way ANOVA followed by Tukey's post hoc test or Kruskal–Wallis test followed by Dunn's test where appropriate ($P < 0.05$). **(I)** Metal content (Fe, Cu, Mn, Zn) in roots of WT, *zip2–1*, and *zip2–1/ZIP2::ZIP2-mCitrine* complementation lines, grown 3 weeks on control agar plates. For each biological replicate, 3–4 plants were pooled (n = 3–4). Different letters indicate statistically significant differences between genotypes using one-way ANOVA followed by Tukey's post hoc test or Kruskal–Wallis test followed by Dunn's test where appropriate ($P < 0.05$).
(TIF)

**S6 Fig. Characterization of *ZIP8* mutants and complementation lines. (A)** Schematic representation of the genomic structure of *AtZIP8* showing the location of the *zip8–1_cr* and *zip8–2_cr* mutations. Black boxes represent exons, the black lines represent introns, the dashed line indicates the promoter region, and the black angled arrow marks the START codon. **(B, C)** Partial *ZIP8* genomic sequence (top) and protein sequence (bottom) for WT, *zip8–1_cr* and *zip8–2_cr* mutants. WT *ZIP8* encodes a 296 aa protein, while *zip8–1_cr* has a theoretical protein of 2 aa and *zip8–2_cr* encodes only 43 aa due to a premature stop codon. **(D)** Phenotype of WT, *zip8–1cr* and *zip8–2_cr* mutants grown for 3 weeks on soil under long-day conditions. Scale bar: 2 cm. **(E)** FW of plants grown as described in panel D. Different letters indicate statistically significant differences between different genotypes using Kruskal–Wallis test followed by non-parametric Tukey test ($P < 0.05$). **(F-G)** Metal content (Fe, Cu, Mn, Zn) in roots of (F) *UBQ10::ZIP8-mCitrine* lines and (G) *35S::IRT1* lines grown for 3 weeks on vertical control agar plates. Four independent T1 plants were pooled to form one biological replicate (n = 3–4). Data are represented as mean ± SD, with statistical differences determined using Student's *t*-test or Mann-Whitney test (*$P < 0.05$; **$P < 0.01$; ***$P < 0.0001$).
(TIF)

**S7 Fig. Genomic structure and phenotype of *zip3* and *zip5* mutants. (A)** Relative *ZIP3* and *ZIP5* transcript levels in WT, *zip3–1*, *zip5–1* and the double mutant *zip3–1zip5–1* grown for 1 week under Zn-deficient conditions. Four biological replicates were analyzed, with each replicate consisting of a pool of at least 40 root seedlings. Data are represented as mean ± SD, with statistical differences between genotypes per individual gene determined using Kruskal–Wallis test followed by non-parametric Tukey's test ($P < 0.05$). **(B, C)** Schematic representation of the genomic structure of *AtZIP3* and *AtZIP5,* showing the location of the T-DNA insertion in *zip3–1*, *zip5–1* and the *zip3–2_cr and zip5-2-2_cr* CRISPR-induced mutations. Black boxes represent exons, the black lines represent introns, the dashed line indicates the promoter region, and the black angled arrow marks the START codon. Partial *ZIP3* and *ZIP5* genomic sequences (top) and protein

sequences (bottom) for WT, $zip3–2_{cr}$ and $zip5–2_{cr}$ mutants. WT ZIP3 encodes a 339 aa protein, while $zip3–2_{cr}$ encodes a truncated protein of 61 aa. Similarly, WT ZIP5 encodes a 360 aa protein, while $zip5–2_{cr}$ produced a truncated protein of 29 aa. **(D)** Representative phenotype of WT, $zip3–1$, $zip3–2cr$, $zip5–1$, $zip5–2cr$ and $zip3–1zip5–1$ plants grown on soil for 3 weeks under long-day conditions. Scale bar: 2 cm. **(E)** Violin plots showing the distribution of the FW of plants grown as described in panel D (n ≥ 10). In the violin plots dashed lines represent the median and dotted lines represent the first and third quartiles. Different letters indicate statistically significant differences between genotypes using one-way ANOVA followed by Tukey's post hoc test ($P < 0.05$). **(F)** Representative images of WT plants, $zip3–1$,$zip5–1$ mutants and the double mutant $zip3–1zip5–1$ mutants grown vertically for 20 days under control conditions (+Zn) and Zn-deficient conditions (-Zn). Plants were directly scanned on their growing plates. Scale bars: 1 cm. **(G-H)** Violin plots showing the distribution of **(G)** FW and **(H)** chlorophyll content of plants grown as described in panel H (n ≥ 10 for I and n ≥ 5 for J). Dashed lines of the violin plots represent the median, while dotted lines mark the first and third quartiles. Different letters indicate statistically significant differences between genotypes, as determined by one-way ANOVA followed by Tukey's post hoc test ($P < 0.05$). (TIF)

**S8 Fig. ZIP3 and ZIP5 individual contribution in metal acquisition. (A)** Zn shoot/root ratio in WT and $zip3–1zip5–1$ grown three weeks on soil under long day conditions. **(B)** Relative transcript levels of 13 *ZIP* members in WT and $zip3–1zip5–1$ roots from plants grown 1 week under -Zn conditions. Each dot represent one biological replicate. Data are represented as mean ± SD (n = 3, each sample is a pool of at least 40 seedlings). Statistical differences were determined using Student's *t*-test or Mann-Whitney test (*$P < 0.05$; **$P < 0.01$; ***$P < 0.0001$). **(C-H)** Metal content (Fe, Cu, Mn, Zn) in shoots **(C, D, E, G)** and roots **(F, H)**. Three to four plants were pooled to form one replicate (n = 4). **(C, D, E, G)** Plants were grown 3 weeks on soil under long-day conditions. **(F, H)** Plants were grown 3 weeks on agar control plate. **(C)** Shoot metal content in WT and $zip3–2_{cr}$ plants. **(D)** Shoot metal content in WT and $zip5–1$ plants. **(E)** Shoot metal content in WT and $zip5–2_{cr}$ plants. **(F)** Root metal content in WT and $zip5–2_{cr}$ plants. **(C-F)** Statistical differences between genotypes for a given element were determined using Student's *t*-test or Mann-Whitney test (***$P < 0.0001$; **$P < 0.01$; *$P < 0.05$). **(G)** Shoot metal content in WT, $zip3–1$ and two complementation lines $zip3–1/ZIP3::ZIP3-mCitrine$ grown 3 weeks on soil. Different letters indicate statistically significant differences between genotypes for a given element using one-way ANOVA followed by Tukey's post hoc test ($P < 0.05$). **(H)** Root metal content in WT, $zip5–1$ and two complementation lines $zip5–1/ZIP5::ZIP5-mCitrine$ grown for 3 weeks on control agar plate. For the complementation lines, 3–4 independent T2 plants were pooled to form one biological replicate. Different letters indicate statistically significant differences between genotypes for a given element using one-way ANOVA followed by Tukey's post hoc test ($P < 0.05$). (TIF)

**S9 Fig. Phenotype of *ZIP::IRT1* lines. (A)** Representative images of WT, $zip3–1zip5–1$ and $zip3–1zip5–1/ZIP_{2/3/5}::IRT1$ lines grown vertically for 10 days under control conditions (+Zn) and Zn-deficient conditions (-Zn). Three representative plants from each genotype were transferred to an agar plate for imaging. Scale bars: 1 cm. **(B)** Violin plots showing the distribution of root length of plants grown as described in panel A (n ≥ 15). Dashed lines of the violin plots represent the median, while dotted lines mark the first and third quartiles. Different letters indicate statistically significant differences between genotypes, as determined by one-way ANOVA followed Kruskal–Wallis test followed by Dunn's test ($P < 0.05$). (TIF)

**S10 Fig. ZIP3 localization depends on Zn availability. (A)** Representative confocal images of Arabidopsis WT root tips stained with propidium iodide (PI). Seedlings were grown on agar plates containing Zn (15 µM) for 5 days, then transferred to liquid medium for 16 h under either Zn-deficient conditions (-Zn) or with a 10-fold excess of Zn (10x Zn; 150 µM) prior to imaging. PI was used to evaluate whether 10xZn conditions induce cell death in the root tip, revealed by staining of the nucleus. PI is shown in gray. Scale bars: 25 µm. **(B)** Representative surface views of epidermal cells of

plants expressing *UBQ10::ZIP3-mCitrine.* Seedlings were grown as in pannel A and transferred to liquid medium for 16 h under either Zn-deficient conditions (-Zn) or with different Zn concentration (15 µM to 150 µM Zn) prior to imaging. The mCitrine signal is shown in Fire (LUT) color scale. Scale bars: 25 µm. **(C)** Quantification of the ratio of plasma membrane (PM) to intracellular (intra) of ZIP3-mCitrine signal (n ≥ 20). Data are represented as mean ± SD, and statistical differences between conditions were determined by one-way ANOVA followed by Tukey's post hoc test ($P < 0.05$). **(D)** Surface views of root epidermal cells from 5-day-old plants expressing *UBQ10::ZIP3-mCitrine* grown on agar plates containing Zn for 5 days, followed by 16 hours in Zn-deficient (-Zn) or Zn-excess (10x Zn) as described in Fig 7. Prior to imaging, plants were treated with 100 µM CHX for 4 hours. The mCitrine signal is shown in Fire (LUT) color scale. Scale bars: 25 µm. (TIF)

**S11 Fig. Map of plasmids. (A)** Map of the modified *pDONRP4-P1R* (Thermo Fisher Scientific) and **(B)** of the modified *pDONR221* (Thermo Fisher Scientific). Maps were generated with Snapgene version 5.0.8. (TIF)

**S1 Table. Mineral content of WT plants grown under +Zn and -Zn conditions.** (XLSX)

**S2 Table. Functional characterization of ZIP fusion proteins.** (XLSX)

**S3 Table. Primer list.** (XLSX)

**S4 Table. Primer efficiency.** (XLSX)

**S5 Table. Clathrin expression.** (XLSX)

**S6 Table. Raw data file.** (XLSX)

## Acknowledgments

We would like to thank Lothar Kalmbach, Satoshi Fujita, Robertas Ursache and Joop E.M. Vermeer for sharing plasmids. We also thanks Léa Jacquier for help to perform statistics in the R environment and Emilie Demarsy for her assistance with chlorophyll quantification and fluorescence measurements as well as Isabelle Fleury for her help in plant propagation. We thank Sylvain Philippe Loubery for help with the microscopes and Anna Puzyrko, Elliot Bowles and Laura Pillard-Tapia for technical assistance during their internships. Special thanks to Sandrine Chay and Stephane Mari from the Multi-Elemental Analyses Service (SAME) at the Institute for Plant Sciences of Montpellier (IPSiM) and advises for Zn uptake experiments. We would like to thank also Léon Dirick for sharing the yeast *fet3fet4* strain. We also extend our thanks to the Photonic Biomaging Center at the University of Geneva.

## Author contributions

**Conceptualization:** Kevin ROBE, Marie Barberon.

**Data curation:** Kevin ROBE.

**Formal analysis:** Kevin ROBE, Marie Barberon.

**Funding acquisition:** Marie Barberon.

**Investigation:** Kevin ROBE.

**Methodology:** Kevin ROBE, Linnka LEFEBVRE-LEGENDRE.

**Project administration:** Marie Barberon.

**Resources:** Kevin ROBE, Linnka LEFEBVRE-LEGENDRE, Fabienne CLEARD.

**Supervision:** Marie Barberon.

**Validation:** Kevin ROBE, Marie Barberon.

**Visualization:** Kevin ROBE.

**Writing – original draft:** Kevin ROBE, Marie Barberon.

**Writing – review & editing:** Kevin ROBE, Marie Barberon.

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
