## [Decision Letter · Decision Letter 0]

PGENETICS-D-25-00240

Four ZIPs contribute to Zn, Fe, Cu and Mn acquisition at the outer root domain

PLOS Genetics

Dear Dr. Barberon,

Thank you for submitting your manuscript to PLOS Genetics. Your study has now been seen by three reviewers. You will see from their comments below that while they find your work of interest, some important points are raised. We remain interested in the possibility of publishing your story in PLOS Genetics, but would like to consider your response to these concerns in the form of a revised manuscript before we make a final decision on publication. We therefore invite you to revise your manuscript taking into account all reviewers comments. 

Please submit your revised manuscript within 60 days Jun 07 2025 11:59PM. If you will need more time than this to complete your revisions, please reply to this message or contact the journal office at plosgenetics@plos.org. Please include the following items when submitting your revised manuscript:

We look forward to receiving your revised manuscript.

Kind regards,

Arun Sampathkumar

Academic Editor

PLOS Genetics

Anne Goriely

Editor-in-Chief

PLOS Genetics

Aimée Dudley

Editor-in-Chief

PLOS Genetics

Anne Goriely

Editor-in-Chief

PLOS Genetics

**Additional Editor Comments (if provided):**

**Journal Requirements:**

**Reviewers' comments:**

Reviewer's Responses to Questions

**Comments to the Authors:**

Reviewer #1: In this study by Robe et al., the authors functionally characterized four ZIP family transporter in Arabidopsis, in terms of expression pattern, tissue localization, metal-dependent phenotypic analysis of knockout lines. Firstly, they found that ZIP2 is a Cu transporter and potentially an Mn transporter, involved in Cu and Mn acquisition from soil. Secondly, they found ZIP8 is an Fe transporter which is polarly localized at the epidermis face to the soil side. Thirdly, they found ZIP3 and ZIP5 function redundantly in the uptake of Zn in the outer root domain, and they also observed the substrate induced endocytosis of ZIP3 in response to Zn excess. The overall presentation of the work is well done, and the study is easy to follow. To enhance the manuscript’s clarity and quality, I suggest addressing the following points.

Major:

1. In figure 1 E, F, is it necessary to present the result of Al? Since Al is mainly active at low PH condition. What is the pH of the culture medium? Actually, I don’t think figure 1 is necessary, as it is commonly Known that Zn deficiency affects the growth of plant and also mineral elements accumulation.

2. The authors demonstrated that ZIP2, ZIP8, and ZIP3, 5 were involved in the uptake of Cu, Fe, and Zn respectively. However, they only showed the mineral element accumulation in roots and shoots. For more direct evidence in element uptake, I suggest using the stable isotopes of these elements for short-term uptake experiments.

3. It appears that the expression of ZIP2, ZIP8, and ZIP3, 5 is induced by Cu, Fe, and Zn deficiency respectively, I recommend to determine the mineral element accumulation under low-concentration treatment condition, which may make the physiological phenotypes clearer.

4. Line 291, for +Fe condition, which form of the Fe was used? Line 298-301, if ZIP8 is involved in Fe uptake, please explain why the accumulation of Fe decreased only in the root of KO lines but not shoot. As I mentioned previously, it is better to perform a short-term uptake experiment by using a stable isotope to directly show the function in uptake.

5. For ZIP3/ZIP5, similar to ZIP8, if ZIP3 and ZIP5 is involved in Zn uptake, please explain or discuss further why Zn accumulation decreased only in root but not in the shoot of the double mutant.

6. The authors stated that by using UBQ10::ZIP3-mCitrine plants, ZIP3 protein levels are affected by high zinc-induced endocytosis. How about under the endogenous promoter condition? Does ZIP3 synthesis itself respond to zinc concentration? If protein synthesis is inherently influenced by zinc concentration (for example, if high Zn significantly inhibits the synthesis of the protein level of ZIP3), then study on protein degradation would become less meaningful.

Minor:

1. Figure 4 E and F, it would be better to show results from two independent knockout lines here.

2. In figure 5 E, why are there two “WT” presented? But in figure 5F, there is only one WT.

3. Line 470-473, if ZIP3 and ZIP5 are not involved in root-to-shoot translocation, may be check the expression of other ZIPs in zip3/zip5 double mutants.

Reviewer #2: The authors provide various types of data for functionally characterizing genes encoding proteins of the ZRT-IRT-related protein family that are candidates for roles in micronutrient metal uptake. The data are partly confirmatory (e.g. contribution of ZIP2 in Cu uptake; Wintz et al., 2003, and others) but add some new or additional data compared to previous publications. All in all, however, the manuscript provides some interesting observations, while the reasoning for narrowing down to specific transporters and some of the conditions used, ..., are somewhat incomplete, sketchy, suboptimal or limited (e.g., protein localization data are only shown for a small set of ZIP proteins and conditions, Fig. 3). The authors should alert readers in the results to the limitations and also discuss and their implications specifically (see below). The suggested contributions of ZIP3 and ZIP5 to Zn uptake supported by the data in this manuscript have recently been published by others (10.1111/tpj.17251), yet additional support remains of scientific value at this point. A (minor, but biologically relevant) contribution of ZIP8 to Fe uptake is novel and was not previously shown.

Major points

1. Line 140-143 and associated data/Figures: There is an issue with the authors‘ data on ZIP transcript levels upregulated in Zn-deficient plants: It is widely known and published by numerous authors that ZIP4 transcript levels (and its promoter activity, based on pGUS lines) are upregulated under Zn deficiency and that ZIP6 transcript levels aren’t. The authors should check their primer sequences and data, and/or repeat their experiments (mixup?). If the data from the authors remain contradictory to all previous publications, they should provide an explanation for this in the discussion, or at least explain the limitations of their data.

2. Line 146 and following: The argument that higher transcript levels of ZIP2 than of other ZIPs under control conditions indicate a potential role in basal metal uptake is not valid in this form. Even lower transcript levels of other ZIPs are compatible with such roles, and it is clear that there is substantial regulation at the protein level in this family (and this manuscript supports this). The authors should modify the text accordingly.

3. Lines 148 to 171: The reviewer sees it as a major limitation of this manuscript that images showing detection and localization of promoter activities for ZIPs are only shown for control conditions (Fig. 2C) and not for Zn deficiency conditions, because growth conditions often result in changes of the extent and domains of promoter activities. Although they later show information for specific deficiency conditions for various individual selected ZIPs, they use the limited data presented in Fig. 2C to select candidates for uptake functions. The authors should therefore highlight the limitations of this approach both here in the results (adjust the reasoning provided to narrow down among ZIPs in this section) and also again in the discussion. The reviewer also feels that this part of the text over-interprets the data shown, and thus should be changed. The authors should also clarify in the caption text of Fig. 2C that seedlings were cultivated in control conditions in order to clarify this.

4. In this context, the conclusion for ZIP4, ZIP9 and ZIP12 promoter activity was minimal (line 161) is not supported by the shown images (Fig. 2C) in the present form, and the fact that these images are from control conditions is decisive and should be mentioned here also, at least. Also the statement in line 166-167 has the limitation of referring to control conditions only. A naive reader would observe activity in epidermal cells also for ZIP4, ZIP9 and ZIP12 (Fig. 2C). The authors don’t argue or provide evidence for disregarding this despite some green signal shown in the epidermal and/or cortex cell layers in Fig. 2C (please consider also the results shown here 10.1111/tpj.17251). Transcript levels in Fig. S1 are consistent with a major role for ZIP9 under zinc deficiency (ZIP4 data are questionable – see above), and also ZIP12 cannot be entirely discounted as its transcript levels are strongly upregulated under Zn deficiency.

5. The reviewer sees the ionome data as another major limitation of this study. First, it was only conducted under one condition and not additionally under a controlled deficiency condition on synthetic media (e.g. Fig. 4E, Fig. 5E, Fig. 6E). Second, the numbers are strange, and this is probably a consequence of using a strangely composed medium. For example, all Cu concentrations in plant tissues are strongly in the deficient range (8 µg g-1 are sufficient, <= 4 deficient; here concentrations are at 1-2). The used media (MS-based) are well-known to be Cu-deficient (e.g. Bernal et al., 2012) and extensively discussed in the literature as highly problematic for metal nutrition research for a variety of reasons. Shoot Zn and Mn levels are very high (10- to 20-fold above normal sufficient levels) and far above nutritional needs in the luxury range, in the control medium. The interpretation of the observations should be adjusted to this explicitly, and this also indicates that all the data of this study were obtained from plants cultivated under conditions of strongly skewed metal homeostasis. Now, some central differences between mutant and the wild type are confirmed in soil, but the authors should generally be very cautious about excluding anything based on their results. Finally, the authors apparently grew plants at high plant density on agar-based media in constant light for as long as 3 weeks. By that time, the plants will be very large, interfering with one another, and they may also have exhausted the nutrients present in the medium of their respective petri plate. So, all in all, these data are not particularly high quality. The authors should mention and discuss all these limitations. They should also add in the methods the information of how many plants were grown per plate, etc. (see also below).

6. Please clarify in the manuscript explicitly which of the mCitrine fusions complement the respective mutants or double mutants (i.e., with data provided in the manuscript), which ones don’t (provide the data if it exists), and for which ones this was not investigated yet. Please discuss this specifically and exhaustively. A supplemental table listing summarizing this might be helpful.

7. Please clarify in results and discussion why we see little visible difference between WT and zip3 zip5 in Fig. 6A (0 µM Zn), whereas 6D suggests an about 50% difference in chlorophyll level.

8. Please clarify in the results and discussion your interpretation of the Zn concentration data shown in Fig. 6E-H: Zn levels are far in the luxury range even in the double mutant. This comment is an example of a specific comment relating to the more general comment above (no. 5).

9. Reviewer suggests to include rice ZIP proteins in the tree in Fig. 2 (or an additional one shown in the supplement), pointing out the ones implicated in Zn uptake.

Minor points

10. Abstract: Suggest to delete “key“ in line 16 (explained elsewhere). Suggest to replace “underscoring the significance“ (line 19) by “in agreement with expectations“. It is also important to clarify in the abstract that statements in line 14 are limited to the conditions employed and the specific line of investigation pursued in this manuscript. Add “plants“ at the end of line 10. Reviewer feels that line 11/12 family name should be in Roman (not italic) fonts.

11. The authors should add a discussion of the somewhat surprising finding that Zinpyr fluorescence (and differences between the wild type and the zip3 zip5 double mutant) were detected only in stele and endodermis (line 354, Fig. 6F and G). The authors report the proteins to be localized in the epidermis primarily, by contrast.

12. It would be nice to see results similar to Fig. 7 also for lower Zn concentrations for readers to receive an idea of the conditions under which this is triggered – 150 µM Zn might be toxic in their medium (maybe the authors could add some marker data to examine this).

13. Line 47: cite also Van der Zaal et al. (1999) on ZAT = MTP1.

14. Line 53: “sativa“

15. Line 92: suggest to replace “primarily involved“ by “contribute“, given the previously published importance of CuII reduction for Cu uptake (emphasizing the roles of COPT proteins) and the major role of IRT1 in Fe uptake.

16. Line 96: replace “emphasize“ by “provides circumstantial support for“ (do this also elsewhere where this argument is repeated)

17. Line 205: “endodermal compartments“: Is this a typo?

18. Line 328: Niehs et al. has not been reviewed. Please refer to at least one peer-reviewed publication where this is shown (e.g. Sinclair et al., 2018).

19. Line 328 “significantly smaller“: All effects described should be significant. It would be far more informative for the readers if the authors provided a quantitative summary of the degree of reduction in growth.

20. Lines 331, 360 “redundantly“: Reviewer suggests the replace by “partially redundantly“. Reviewer believes that the data provided in this manuscript rather suggest compensatory effects in mutants, and for now we do not know much about whether there is redundancy among ZIP3 and ZIP5 in wild-type plants.

21. Line 431-433: Remove “we identified“ – other authors have published quite a bit of data on that. The authors provide reverse genetic evidence for this here.

22. Line 339: The authors should specify the nutrients addressed in this cited study and also mention the well established knowledge on the central importance of root-pressure/xylem loading dependent flux in the xylem.

23. Line 459 “are key transporters“: Replace by “contribute to“. The phenotypes are not severe. It is quite possible, especially given the limitations of this study, that other transporters contribute to this, a conclusion that would also align with 10.1111/tpj.17251 (see also 10. above).

24. Lines 531-532: Include full description or a reference to a publication where this is fully described.

25. Lines 538-539: Please modify Fig. S2 to clarify the position of mCitrine, label TMH by numbers and clarify cytosolic side.

26. Lines 578: For all experiments on plates, add the number of seedlings per plate.

27. Describe in methods the harvesting of materials from soil-grown plants, especially the harvesting and cleaning of roots for multi-element analysis (e.g. Fig. S7E).

28. Line 580/81: Please clarify how these plants were grown, including number of seedlings per plate.

29. qRT-PCR lines 617-626. Please clarify the number of seedlings per plate and growth conditions. Please provide data on Clathrin control gene expression under the conditions employed here. Please provide data for primer pair amplification efficiencies.

30. Please define what is meant by “biological replicates” in Line 626 and wherever else this term is mentioned in the manuscript – grown alongside on the same plate, on a different plate, or grown at a separate time (independent experiment?),...

31. Line 630: Please add the pH of the EDTA solution.

32. ICP-OES: Please add instrument name/manufacturer, wavelengths, information on calibration (frequency) and QC.

33. Fig. 1D: Reviewer cannot find dashed line in the orange violin shape. Please check that all others are complete.

34. Fig. 4E: Change the use of axis breaks so that also Fe concentrations can be deciphered.

35. Fig. 5E: Please modify legend to explain the two WT in this figure.

36. All vertical axis titles for root length: Please correct typo.

37. Why are Zn and Mn levels in Fig. S4I lower than elsewhere? Why are they lower in Table S1? What was different from the other experiments? Please specify in methods and in the caption texts.

38. It would be nice to see root data for Fig. S6 or S7 or another experiment in which Zn supply was not as excessive as on the agar-based media, also for zip3 (not only for zip5) and the zip3 zip5 double mutant.

39. All experiments: The authors should provide ZT of harvest for all experiments involving plant growth in light-dark cycles.

Reviewer #3: Dear authors

The paper presents new data regarding as least two standing questions in the field: whether ZIP2 indeed transports copper, and which ZIPs are the primary zinc transporters for zinc uptake in arabidopsis. The data is clear and of high quality. My only major concern is with text organization: authors need to carefully separate sections in paragraphs, since many times the text is continuous even when topics change, including the introduction and results sections. Moreover, I suggest authors being more upfront about the recently published paper by Ochoa Tufiño in Plant Journal, which also demonstrates that ZIP3 and ZIP5 are zinc uptake transporters. Authors could compare and let readers know what is consistent, making a synthesis of both datasets. I believe this would not hurt the "novelty factor" of this work, but rather would make it even more valuable.

Here are my other minor comments:

• Line 31: I think you can cite only as Palmer and Guerinot 2009?

• Line 43-44: “Cation Diffusion Facilitator (CDF) family, and the Metal Tolerance Proteins (MTP) family”. I think authors can mention only one of these families, as the CDFs are named MTPs in plants (i.e., they are not two distinct gene families as implied by the phrase).

• Line 53: please correct to “Oryza sativa” (sativa should be minor key, nor Sativa).

• Line 66-67: “Although AtIRT3 overexpression results in Zn and Fe accumulation” – please provide the reference for this.

• The paragraph from line 41 to 86 is very long and jumps from one topic to another. It needs careful reorganization. I suggest to make two paragraphs out of it; one more general about the ZIPs, an another about ZIP and Zn transports, mentioning the recently discovered function of OsZIP5/OsZIP9.

• Line 108: I suggest changing “the plants stunted growth and chlorosis” to “the plants showed stunted growth and chlorosis” or similar.

• Line 138: I am aware that the authors briefly mention the recently published paper bu Ochoa Tufino in Plant Journal in their discussion. I also understand that this paper came out before and authors only became aware of it after finishing their own work. However, I think the community would benefit from a comparison of gene expression results of both papers. Authors should consider performing that, so readers can see what is consistent.

• Line 149: this could the start of a new paragraph, to improve the flow of the text: “Anticipating that transporters…”

• Line 169-170: I understand that this is possible, but it’s highly speculative based solely on this data. I suggest suppressing it here and moving to the discussion sections, making sure to highlight that it’s only a speculation.

• Line 170: “Based on our expression analysis” could be the start of a new paragraph. Also, I think it’s “analyses”, since you performed several, not only one.

• General comment: the text needs structural revisions, since often there is not paragraph separation. Revise throughout.

• Line 289: “both zip8 alleles” sounds like colloquial lab language. We use it, but in the paper I suggest using “both zip8 mutant lines”.

• Line 309: I think the claim “…does not fully overlap with IRT1’s function” is not supported by the evidence. Although compelling to claim that ZIP8 is indeed involved somehow in Fe acquisition, whether that overlaps with IRT1 would need double mutants and gene swap experiments, for example. I think authors should suppress it, or move it to the discussion section.

• Figure 6 and data therein: could authors provide a clearer/closer image of plant shoots under +Zn and -Zn. Despite the chlorophyll measurement, it would be key to see what is the visual phenotype I shoots of the double mutant zip3zip5 under -Zn. Since this is one of the key finding of the paper, I strongly suggest authors to provide that.

• I suggest inverting figure 6G to match the common orientation used in the paper.

• Line 333-335: why there is not ICP data for the single mutants?

• In this section of the paper the lack of proper organization in paragraphs is very clear. Authors need to carefully revise that. It does not affect the content quality of the manuscript, but should be reorganized.

• Line 345: “shoot analysis revealed that ZIP3 is primary responsible”. This is based on the observation that zip3 mutants have lower Zn concentration? I think authors could be a bit less strong about it, since this suggest ZIP3 has a primary role over ZIP5.

• Why the ICP data on singles and doubles not presente together. I think it should be in the main figures.

• Line 351: “To gain further insight...” is a clear new paragraph, as an example to help authors organize the text.

• Line 463-464: please correct to “Zn uptake IN root”.

**Have all data underlying the figures and results presented in the manuscript been provided?**

Reviewer #1: None

Reviewer #2: **No: **

Reviewer #3: Yes

PLOS authors have the option to publish the peer review history of their article (what does this mean? ). If published, this will include your full peer review and any attached files.

**Do you want your identity to be public for this peer review?** For information about this choice, including consent withdrawal, please see our Privacy Policy .

Reviewer #1: No

Reviewer #2: **Yes: ** Ute Krämer

Reviewer #3: **Yes: ** Felipe Klein Ricachenevsky

**Figure resubmission:**
---

## [Decision Letter · Decision Letter 1]

PGENETICS-D-25-00240R1

Four ZIPs contribute to Zn, Fe, Cu and Mn acquisition at the outer root domain

PLOS Genetics

Dear Dr. Barberon,

Many thanks for submitting the revised manuscript "Four ZIPs contribute to Zn, Fe, Cu and Mn acquisition at the outer root domain". As you will see, all three reviewer find the manuscript is improved, with the earlier comments being addressed. There are some minor points raised by reviewer two, that mainly requires modification to certain parts of the text. In light of these reports, I am glad to inform you that your manuscript is provisionally accepted for publication in PLOS Genetics, pending the minor comments that needs to be fixed.

Please submit your revised manuscript within 30 days Jul 24 2025 11:59PM. If you will need more time than this to complete your revisions, please reply to this message or contact the journal office at plosgenetics@plos.org. Please include the following items when submitting your revised manuscript:

We look forward to receiving your revised manuscript.

Kind regards,

Arun Sampathkumar

Academic Editor

PLOS Genetics

Anne Goriely

Editor-in-Chief

PLOS Genetics

Aimée Dudley

Editor-in-Chief

PLOS Genetics

Anne Goriely

Editor-in-Chief

PLOS Genetics

**Journal Requirements:**

**Reviewers' comments:**

Reviewer's Responses to Questions

**Comments to the Authors:**

Reviewer #1: The revised version of the manuscript shows significant improvement compared to the previous submission. The authors have carefully addressed all the major concerns I raised in my initial review. The clarification and additional data provided have strengthened the overall quality and clarify the manuscript. I have no additional comments at this time.

Reviewer #2: In their first revision, the authors have adequately addressed most of the issues I had raised. Only a few minor aspects are remaining and need some additional attention or modifications (see below).

1. Fig. 1B labeling Log10(fold change -Zn vs. + Zn), and this text would be better placed next to the legend relating numbers to colors (or explain in caption text). After describing the results shown in this panel in the results text, the authors should cite the publications in which the upregulation of transcript levels of each of the respective ZIPs under Zn deficiency was first reported in roots of Arabidopsis thaliana (e.g. partly Talke et al., 2006; Assuncao et al., 2010; Sinclair et al., 2018, ..., for example).

2. Caption text Fig. 1C: “a single focal plane”; please add here that seedlings were grown under control conditions (age would be useful, too).

3. Fig. 3A: The left panel says –Cu, but most likely the agar contained some Cu so that the total Cu level will not be zero. The authors should make this more explicit somewhere (methods).

4. Fig. 4, caption text, final line “obtained”.

5. L. 68: “vacuolar Zn storage”

6. L. 300: suggest to replace “critical role for ZIP2” by “contribution of ZP to”, given the extremely low Cu levels even in wild-type in plants under the –Cu conditions used here and the quantitatively only small relate difference between zip2 mutants and the wild type.

7. Line 309: Similarly (see 9.), suggest to delete “major” in line 309.

8. Line 317: Suggest to replace “in the root” by “into the root”

9. Line 335: “ a truncated peptide of 2 aa in length”; add length of predicted produced peptide also for zip8-2.

10. Line 351 “of both zip8 mutant lines”

11. Line 406: On agar plates, rosette tissues may absorb or take up metals from the medium. This may explain the comparably high metal levels in above-ground tissues, and it is not possible in soil-grown plants. This could also be added when discussing these observations.

12. Lines 413-417: Given that root-to-shoot Zn translocation ratios are similar in zip3 zip5 and WT (Fig. S8A), the reviewer is not sure about the role proposed here of other ZIPs in root-to-shoot Zn translocation in the zip3 zip5 double mutant. The authors explain their thoughts a little better in the discussion lines 608-611, and maybe this text should be moved there from the results.

13. Line 423: “reduction” reads like efflux of Zn from the roots of the double mutant, which is not what the authors mean here – it would be better to rephrase this sentence.

14. Line 569: The authors should specify what they mean by “this function” here.

15. Line 575: “are closer to”. The authors should write more precisely what they mean by this.

16. Line 594: “in this point of view” – the authors should state more precisely what they mean by this (e.g. which point of view).

17. There are lacking spaces in some positions in the manuscript text, and also some grammatical errors, typos, formatting issues (references) and stylistic issues (e.g. first words identical in lines 612 and 620), especially in the methods.

18. Line 634: The reviewer feels that this text needs a few additions. The authors should note that MTP2 transcript is only present in highly Zn-deficient plants, and a symplastic transport of Zn2+ from cell to cell via plasmodesmata is not only possible via the ER but also within the cytosolic regions surrounding the desmotubule. The authors should also consider that Zn transport could be entirely symplastic after uptake in the outer cell layers, and additional Zn uptake could occur into endodermal cells without any Zn efflux on the inside of cortex cells (the presence of an uptake system on the endodermis cannot provide evidence for efflux from outer cell layers of the root).

19. Methods: Precise PCR conditions and other methodology for obtaining DNA fragments, cloning procedures with appropriate literature references (including also the origin of the mCitrine) should be specified in detail.

20. Methods: The authors should specify the type (shape, dimensions) of agar plates used for plant cultivation and the volume of medium per plate.

21. Methods: It appears that many experiments were done in T1. The authors should specify how and when they selected transgenics in the experiments described in lines 725-752.

22. Lines 792-3: Sentence needs fixing (redundancies).

23. Statistics: It would be good to know how many independent replicate agar plates were reflected (approximately) among all biological replicates.

24. Authors should indicate at which stage plasmids, constructs or inserts were sequenced to verify, the extent, and the technique used.

Reviewer #3: The authors have adequately improved the manuscript.

**Have all data underlying the figures and results presented in the manuscript been provided?**

Reviewer #1: None

Reviewer #2: **No: ** It would be nice if the authors could provide (submit to a database) the full sequences of their inserts (at least) or better even the full plasmids employed.

Reviewer #3: Yes

PLOS authors have the option to publish the peer review history of their article (what does this mean? ). If published, this will include your full peer review and any attached files.

**Do you want your identity to be public for this peer review?** For information about this choice, including consent withdrawal, please see our Privacy Policy .

Reviewer #1: No

Reviewer #2: **Yes: ** Ute Kraemer

Reviewer #3: **Yes: ** Felipe Ricachenevsky

**Figure resubmission:**
---

## [Editor Report · Decision Letter 2]

Dear Dr Barberon,

We are pleased to inform you that your manuscript entitled "Four ZIPs contribute to Zn, Fe, Cu and Mn acquisition at the outer root domain" has been editorially accepted for publication in PLOS Genetics. Congratulations!

Yours sincerely,

Arun Sampathkumar

Academic Editor

PLOS Genetics

Anne Goriely

Editor-in-Chief

PLOS Genetics

Aimée Dudley

Editor-in-Chief

PLOS Genetics

Anne Goriely

Editor-in-Chief

PLOS Genetics

Comments from the reviewers (if applicable):

**Data Deposition**

http://datadryad.org/submit?journalID=pgenetics&manu=PGENETICS-D-25-00240R2

**Press Queries**

---

## [Editor Report · Acceptance letter]

PGENETICS-D-25-00240R2

Four ZIPs contribute to Zn, Fe, Cu and Mn acquisition at the outer root domain

Dear Dr Barberon,

We are pleased to inform you that your manuscript entitled "Four ZIPs contribute to Zn, Fe, Cu and Mn acquisition at the outer root domain" has been formally accepted for publication in PLOS Genetics! Your manuscript is now with our production department and you will be notified of the publication date in due course.

With kind regards,

Anita Estes

PLOS Genetics

On behalf of:
